# Extended Deep Submodular Functions

**Seyed Mohammad Hosseini\***                                    *semo.hosseini@sharif.edu*
*Department of Computer Engineering*
*Sharif University of Technology*

**Arash Jamshidi\***                                              *arashjamshidi@sharif.edu*
*Department of Computer Engineering*
*Sharif University of Technology*

**Seyed Mahdi Noormousavi**                                  *mahdinoormousavi75@sharif.edu*
*Department of Computer Engineering*
*Sharif University of Technology*

**Mahdi Jafari Siavoshani**                                       *mjafari@sharif.edu*
*Department of Computer Engineering*
*Sharif University of Technology*

**Naeimeh Omidvar**                                              *n.o.omidvar@gmail.com*
*Tehran Institute for Advanced Studies (TEIAS), Khatam University, Tehran, Iran*

**Reviewed on OpenReview:** *https://openreview.net/forum?id=qoxusn9u2L*
*\* Equal contribution*

## Abstract

We introduce a novel representation of monotone set functions called Extended Deep Sub-modular functions (EDSFs), which are neural network-representable. EDSFs serve as an extension of Deep Submodular Functions (DSFs), inheriting crucial properties from DSFs while addressing innate limitations. It is known that DSFs can represent a limiting subset of submodular functions. In contrast, we establish that EDSFs possess the capability to represent all monotone submodular functions, a notable enhancement compared to DSFs. Furthermore, our findings demonstrate that EDSFs can represent any monotone set function, indicating the family of EDSFs is equivalent to the family of all monotone set functions. Additionally, we prove that EDSFs maintain the concavity inherent in DSFs when the components of the input vector are non-negative real numbers—an essential feature in certain combinatorial optimization problems. Through extensive experiments, we demonstrate that EDSFs exhibit significantly lower empirical generalization error in representing and learning coverage and cut functions compared to existing baselines, such as DSFs, Deep Sets, and Set Transformers.

## 1 Introduction

Submodular functions have found extensive applications in various fields of study, including modeling influence in social networks Kempe et al. (2003), energy functions in probabilistic models Gillenwater et al. (2012), and clustering Narasimhan et al. (2005). In particular in economics, a wide range of scenarios incorporate the concept of diminishing marginal return, where acquiring more goods results in diminishing the overall satisfaction or the so-called "utility" McLaughlin & Chernew (2001); Kimball et al. (2024).

There are several challenges associated with the widespread use of these functions in recent machine learning applications. To improve the modeling of submodular functions in these applications, efforts have been made to represent submodular functions using differentiable functions, such as neural networks. This representation

enables the solution of key submodular optimization problems through the utilization of gradient-based methods and convex optimization. To illustrate this, suppose we want to maximize a submodular function $f$ under certain constraints. In the exhaustive-search approach, we would have to check all $2^n$ subsets, which is exponential in the size of the ground set $S$. However, with a differentiable representation of $f$, we can utilize first-order information of the function, such as gradients, and project onto the constraint set to solve the optimization problem more efficiently.

There have been previous efforts to represent submodular functions using neural networks. For instance, in Bilmes & Bai (2017), the authors introduced a neural network architecture called Deep Submodular Functions (DSFs), consisting of feedforward layers with non-negative weights and normalized non-decreasing concave activation functions. Functions in this class exhibit interesting properties, such as the concavity of the function when the components of the input vector are all non-negative real numbers. By increasing the number of layers in the DSFs architecture, the family expands, indicating that there are functions in DSFs with $n + 1$ layers that are not present in DSFs with $n$ layers. However, as stated by the authors, DSFs cannot represent all monotone submodular functions, which highly restricts their applicability to many machine learning problems.

In this paper, we introduce a novel neural network architecture, called Extended Deep Submodular Functions (EDSFs), which not only have the capability to represent any monotone submodular functions but can also represent any monotone set functions. Moreover, same as in DSFs, when the components of the input vector are all non-negative real numbers, EDSFs are concave, an important feature applicable in various combinatorial optimization settings. In addition, our experiments demonstrate that EDSFs are able to learn one of the most complicated monotone submodular functions, i.e., coverage functions, with significantly lower empirical generalization error compared to DSFs. We define EDSFs as the minimum of $r$ DSFs. Although in our proofs, exponential number of DSFs are needed to represent any monotone submodular function, in our experiments we observed that we can learn coverage functions and monotone cut functions using much fewer DSFs[1].

The rest of the paper is organized as follows: In Section 2, we formally define DSFs and state some of their important properties. In Section 3, we introduce our augmented architecture to represent all monotone (submodular) set functions and provide a proof. In Section 4, we demonstrate the superior performance of EDSFs in learning coverage functions through numerical evaluations.

## 1.1 Related Works

The exploration of neural networks for modeling submodular functions is relatively sparse in the existing literature. A notable contribution is the introduction of DSFs Bilmes & Bai (2017); Dolhansky & Bilmes (2016). Building on this work, the authors in Bai et al. (2018) address the maximization of DSFs under matroid constraints, using gradient-based methods to solve the optimization problem. Their work provides theoretical guarantees, establishing a suitable approximation factor given the problem's constraints. More recently, a novel architectural approach has been proposed in De & Chakrabarti (2022), which not only preserves submodularity but also extends its applicability to a more generalized form, accommodating $\alpha$-submodular functions. This signifies a notable advancement in the landscape of neural network-based modeling of submodular functions, expanding the scope of potential applications and providing a platform for exploring more nuanced and versatile representations within this domain. There are other works that attempt to represent monotone set functions using neural networks. For example, in Weissteiner et al. (2021), the authors introduce a neural network architecture that represents all monotone set functions and explore their use in designing auctions.

Authors in Zaheer et al. (2017) introduces Deepsets, a method for learning permutation-invariant set functions, offering a flexible architecture that can handle set-based tasks. While their work provides a powerful way to model general set functions, they do not explicitly address the estimation of submodular functions, which require handling the diminishing returns property. This limits its direct applicability to submodular function estimation, as their focus is more on tasks like point cloud classification and set expansion rather than optimizing or estimating submodular structures. Inspired by this work, Lee et al. (2019) introduces the Set

---

[1]All of the codes associated with experiments are available at https://github.com/semohosseini/comb-auction

Transformer, an attention-based neural network architecture designed for processing set-structured data. While the authors demonstrate its effectiveness on various set-input tasks, its performance was not specifically evaluated on estimating set functions such as submodular functions. Wagstaff et al. (2019) talks more about the theoretical side of works like Deepsets and Set Transformer, providing insights into the limitations of representing permutation-invariant functions on sets. While their work doesn't specifically address submodular functions, it provides a general framework for understanding the representational power of neural networks on set-structured data. However, their focus is primarily on theoretical limitations rather than practical algorithms for learning specific classes of set functions, leaving open questions about how these insights translate to the efficient learning of submodular functions in particular.

One of the works related to learning general submodular functions is Feldman & Vondrák (2016). The authors prove tight bounds on approximating submodular functions by juntas, showing that any submodular function can be $\epsilon$-approximated by an $O(\frac{1}{\epsilon^2} \log \frac{1}{\epsilon})$-junta. Although their work gives a good insight into learning submodular functions, it mainly focuses on the theoretical existence of these estimators and doesn't fully provide a practical way of learning them. The authors in Balcan & Harvey (2018) investigate submodular functions from a learning theory perspective, developing algorithms for learning these functions and establishing lower bounds on their learnability. Moreover, the authors in Feldman & Kothari (2014) attempt to approximate and learn coverage functions in polynomial time.

There are some works that talk about the combinatorial and submodular optimization. For example, Sakaue (2021) introduces a differentiable version of the greedy algorithm for monotone submodular maximization, called Smoothed Greedy. One of the strengths of this work is that it maintains theoretical guarantees similar to the original greedy algorithm Vondrak (2008) while enabling gradient-based learning. However, they work doesn't directly address the estimation of submodular functions and only focusing on maximizing a (given) single submodular function. In our work we aim to estimate any given submodular function and use this estimation for downstream tasks related to submodular maximization such as social welfare maximization. in another work, Wilder et al. (2019) introduces a framework called 'decision-focused learning' that aims to bridge the gap between predictive models and combinatorial optimization. Their approach involves end-to-end training of machine learning models to directly optimize decision quality, rather than prediction accuracy. However, in their work, the target function can be described by an unknown parameter and the aim is to perform optimization and learning together, whereas our work aims to learn the submodular function itself from data.

Lin & Bilmes (2012) introduces the concept of submodular shell mixtures for learning submodular functions in a structured prediction setting. While proposed method achieves strong results in document summarization, it reliance on predefined shell and approximate inference may restrict the class of learnable functions. These limitations highlight the ongoing challenges in developing versatile methods for learning submodular functions.

## 2 Background

For a ground set $S$, any function $f : 2^S \to \mathbb{R}$ is referred to as a set function. Below, we present the definitions of essential concepts needed for the remainder of the discussion.

**Definition 2.1.** (Monotone Set Function) A set function $f : 2^S \to \mathbb{R}^+$ is called a monotone set function if for any $A \subseteq B \subseteq S$ we have,

$$f(A) \leq f(B). \tag{1}$$

**Definition 2.2.** (Normalized Set Function) A set function $f : 2^S \to \mathbb{R}^+$ is called a normalized set function if we have,

$$f(\emptyset) = 0. \tag{2}$$

**Definition 2.3.** (Modular Function) A function $m : 2^S \to \mathbb{R}^+$ is called a modular function if we have,

$$\forall A \subseteq S : m(A) = \sum_{a \in A} m(a). \tag{3}$$

Now, we can formally define submodular functions as follows.

**Definition 2.4.** (Submodular Function) A function $f : 2^S \to \mathbb{R}$, where $S$ is a finite set, is *submodular* if for any $A \subseteq B \subseteq S$ and $v \notin B$, we have,

$$f(A \cup v) - f(A) \geq f(B \cup v) - f(B). \tag{4}$$

In the remainder of the paper, without loss of generality, we will focus on normalized monotone set/submodular functions. If the function is not normalized, we can simply subtract the value of $f(\emptyset)$ from the function so as to make it normalized. Note that this transformation also maintains the submodularity of the function.

**Definition 2.5.** (Sum of Concave Composed with Modular Functions Bilmes & Bai (2017)) Assume a finite set $S$ (*nodes* or *input features*) with cardinality $n$. Given a set of $m_1, m_2, \ldots, m_k$ ($m_i : 2^S \to \mathbb{R}^+$) modular functions and $\phi_1, \phi_2, \ldots, \phi_k$ ($\phi_i : \mathbb{R}^+ \to \mathbb{R}^+$) being their corresponding **non-negative**, non-decreasing, normalized (i.e., $\phi_i(0) = 0, \forall i$), concave functions, and an arbitrary modular function $m_\pm : 2^S \to \mathbb{R}$, the *SCMM*, $g : 2^S \to \mathbb{R}$, derived by these functions is defined as,

$$g(A) = \sum_{i=1}^{k} \phi_i(m_i(A)) + m_\pm(A). \tag{5}$$

In the rest of the paper, since we only consider the monotone set functions, we pull our attention into "*monotone*" SCMMs, which the range of modular functions is only the positive real values, namely, $m_\pm : 2^S \to \mathbb{R}^+$.

Based upon recent developments, it has been discovered that the aforementioned functions given in Equation 5 exhibit inherent submodular characteristics Bilmes & Bai (2017). Looking ahead, we can view these functions as a single-layer neural networks equipped with nonlinear activation functions that exhibit concavity properties, similar to a linear mixture of inputs followed by a stepwise activation function such as ReLU (Rectified Linear Unit) after computing the overall output values.

By employing this insight, we can expand the horizons of submodular functions by leveraging them across multiple tiers. We now introduce a pivotal concept that facilitates the utilization of advanced artificial intelligence tools, specifically deep learning.

**Definition 2.6.** (Deep Submodular Function Bilmes & Bai (2017)[2]) Assume we intend to define $L$ (i.e., depth of the network) layer DSF. If $L = 1$ we use an SCMM (without last summation). Therefore, in this case we have a single-layer $n \times k$ network (with *non-negative* weights) with submodular outputs. For $L = l > 1$ we first assume the output of $l - 1$ layers as a given $n \times k$ DSF. Let's denote the output nodes of the given DSF by $B = \{\varphi_1, \varphi_2, \ldots, \varphi_k\}$. Now we want to add one layer at the end of the given DSF. We append a $k \times m$ fully connected layer with *non-negative* weights and corresponding normalized non-decreasing concave functions $\phi_1, \phi_2, \ldots, \phi_m$ for each newly added node. Then we define the outputs of the network as $C = \{\psi_1, \psi_2, \ldots, \psi_m\}$

$$\forall v \in C : \psi_v = \phi_v \left( \sum_{u \in [k]} w_{uv} \varphi_u \right) + b_v, \tag{6}$$

and $b_v \in \mathbb{R}^+$ is a bias parameter of the node. In this scheme, we have new layer added to the DSF. Therefore, we introduced a $n \times m$ DSF with $L = l$ layers. As an example, a 3-layer DSF is shown in Figure 1.

With this understanding, one can guarantee that the outcome of this architecture forms a submodular function, as it is shown in Bilmes & Bai (2017). An established finding regarding SCMMs states that any SCMM employing an arbitrary activation function can be represented as a two-layer SCMM with only the min activation function Bilmes & Bai (2017).

Despite these findings, there exist notable limitations when it comes to DSFs. While DSFs possess a range of capabilities, they are unable to encompass all submodular functions. This implies that there will always be submodular functions that cannot be represented using any number of layers in DSFs Bilmes & Bai (2017). To address this limitation, in Section 3, we extend DSFs by adding a limited number of components in the network architecture to represent all submodular functions.

---

[2]Here, we only focus on the monotone DSFs, so we have omitted the modular term in the original definition of DSFs.

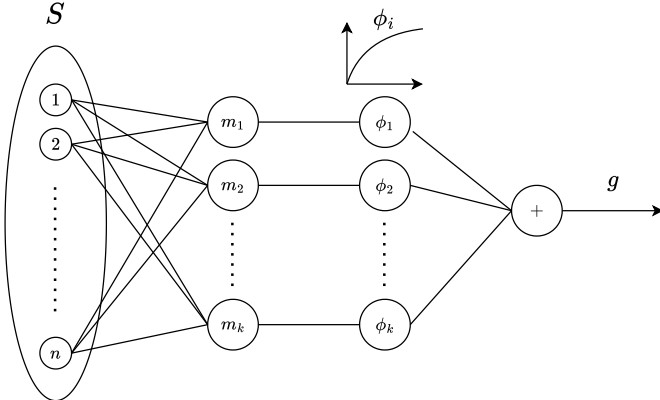

Figure 1: SCMM Architecture. Input vector of the network is a 0-1 vector representing the corresponding subset of the ground set $S$.

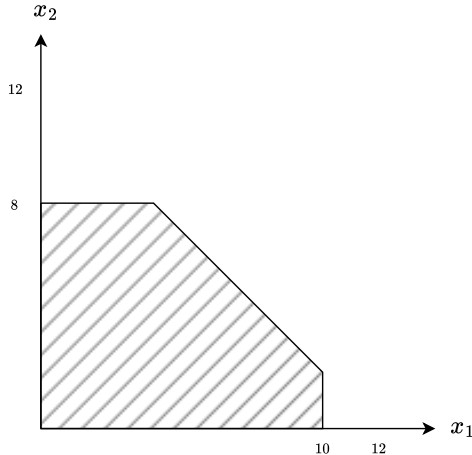

Figure 2: Polymatroid corresponding to the example function with a ground set of size 2.

In the following, we point out certain fundamental concepts in combinatorial optimization that will be employed throughout the remainder of the paper. We commence with an exploration of polymatroids.

**Definition 2.7.** (Polymatroid) Consider a finite set $S$ with $|S| = s$ and a submodular function $f : 2^S \to \mathbb{R}$ defined on $S$. A polymatroid corresponding to $f$, denoted by $\mathcal{P}_f$, is defined as,

$$\mathcal{P}_f = \left\{ \mathbf{x} \in \mathbb{R}^s : \forall A \subseteq S : \mathbf{x}(A) \leq f(A), \mathbf{x} \succeq 0 \right\}, \tag{7}$$

where $\mathbf{x}(A) \coloneqq \sum_{i \in A} x_i$.

*Example* 2.8. As an example of polymatroids, we can consider the set function $f : 2^{\{1,2\}} \to \mathbb{R}^+$ where $f(\{1\}) = 10$, $f(\{2\}) = 8$, and $f(\{1,2\}) = 12$. which the corresponding polymatroid is shown in Figure 2.

The following lemma describes an important property of the polymatroids corresponding to submodular functions.

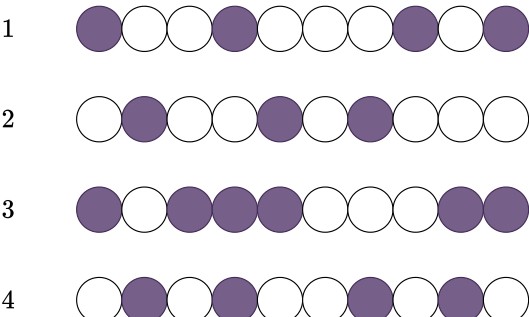

Figure 3: Example of coverage function with universe size 10, ground set of size 4, and with uniform weight of 1.

**Lemma 2.9.** *(Goemans et al. (2009)) For a polymatroid $\mathcal{P}_f$ corresponding to a monotone submodular function $f$, we have*

$$\forall A \subseteq S : f(A) = \sup_{\mathbf{x} \in \mathcal{P}_f} \mathbf{x}(A). \tag{8}$$

We also define the following important submodular problem which we need later in our experiments.

**Definition 2.10.** (Submodular Welfare Maximization) For a collection of $n$ users with $v_1, v_2, \ldots, v_n : 2^S \to \mathbb{R}$ as their (estimated) *submodular* valuation functions on each subset of the finite set $S$ with $|S| = s$, the submodular welfare maximization problem aims to maximize the social welfare function, i.e., the sum of all valuation functions when we *partitioned* set of items $S$ and assigned to the users, and is formally defined as

$$\begin{aligned} \max_{S_i, i=1,\ldots n} \quad & \sum_{i=1}^{n} v_i(S_i), \\ s.t. \quad & S_i \subseteq S, \\ & \bigcup_{i=1}^{n} S_i = S, \quad \forall i,j : S_i \cap S_j = \emptyset. \end{aligned} \tag{9}$$

Moreover, for any partition $A = \{S_1, S_2, \ldots, S_n\}$ of the ground set $S$ to users, we define its efficiency as,

$$\text{Eff}(A) \coloneqq \frac{\sum_i v_i(S_i)}{\sum_i v_i(S_i^*)} \tag{10}$$

where $OPT \coloneqq \{S_1^*, S_2^*, \ldots, S_n^*\}$ is the optimal partition.

Furthermore, in our experimental results, we will employ Coverage Functions, defined as follows.

**Definition 2.11.** (Coverage Function) A function $c : 2^{[n]} \to \mathbb{R}^+$ is a coverage function, if there exists a universe $U$ with non-negative weights $w(u)$ for each $u \in U$ and subsets $A_1, A_2, \ldots, A_n$ of $U$ such that for any $B \subseteq [n]$ we have $c(B) = \sum_{u \in \cup_{i \in B} A_i} w(u)$.

*Example* 2.12. As an example of the coverage function, for a ground set of size 4, we illustrate the coverage function with universe size 10 and with uniform weight of 1 for all of the elements in the Figure 3. In this example we have $f(\{1,3\}) = 7$ and $f\{3,4\}) = 8$.

Alternatively, coverage functions can be described as non-negative linear combinations of monotone disjunctions. There are a natural subclass of submodular functions and arise in a number of applications Feldman & Kothari (2014).

## 3 Extended Deep Submodular Functions

### 3.1 Architecture Definition

Formally, we define Extended Deep Submodular Function as follows.

**Definition 3.1.** A set function $h$ is an EDSF if it can be represented as the minimum of $r$ Deep Submodular Functions $f_1, f_2, \ldots, f_r$, where $r$ is an arbitrary number, namely,

$$h(A) = \min\left\{f_1(A), f_2(A), \ldots, f_r(A)\right\}, \quad \forall A \subseteq S. \tag{11}$$

In the rest of this section, we aim to demonstrate that we can represent any monotone submodular function $f$ using some $g \in$ EDSFs. The idea behind the proof is to leverage the relationship between the minimum of submodular functions and the intersection of their polymatroids. The main result of this paper is summarized in the following theorem.

**Theorem 3.2.** *Family of monotone set functions is exactly equal to the family of Extended Deep Submodular functions (EDSFs).*

In the following, we will go to prove this result step by step. First, we assume that a submodular function $f : 2^S \to \mathbb{R}$ is given. For convenience, we use these notations to simplify our discussion.

**Definition 3.3.** We define:

1. $c_A = f(A)$, and for all $j \in S$ we denote $c_j = f(\{j\})$.

2. a polytope $L_A$ for all $A \subseteq S$ as

$$L_A = \left\{\mathbf{x} \in \mathbb{R}^n : \mathbf{x} \succeq 0, \mathbf{x}(A) \leq c_A, \forall j \notin A : x_j \leq c_j\right\}. \tag{12}$$

3. for any $A \subseteq S$ and $B \subseteq S$ the submodular function

$$g_A(B) = \min\left\{\sum_{j \in A \cap B} w_j, c_A\right\} + \sum_{k \in B \setminus A} w_k, \tag{13}$$

where $w_j = c_A$ for any $j \in A$ and $w_j = c_j$ for any $j \notin A$, and $w_j$ represents the corresponding weight of some neural network.

Using these definitions, in the first step, we will show that each $g_A$ can be represented using a simply two-layer DSF.

**Lemma 3.4.** *For any $A \subseteq S$, we can design a two-layer DSF that represents the $g_A$ function, therefore the $g_A$ function is a submodular function.*

*Proof.* To represent the function $g_A$ using a DSF, we have constructed a network architecture as illustrated in Figure 4. This architecture comprises only two layers. The first layer contains two nodes, one of which employs the minimum function as its activation layer. The second layer consists of a single output node responsible for computing the sum of the inputs.

All edges connected to the lower node possess a weight of $c_A$. Conversely, each edge linked to the upper node bears a weight corresponding to the value $c_j$, specifically $f(j)$ according to the definition.

In this manner, we have introduced the network as a two-layer DSF

$$g_A(B) = \min\left\{\sum_{j \in A \cap B} w_j, c_A\right\} + \sum_{k \in B \setminus A} w_k \tag{14}$$

With this, we conclude our proof. $\qquad \square$

We next introduce a useful lemma.

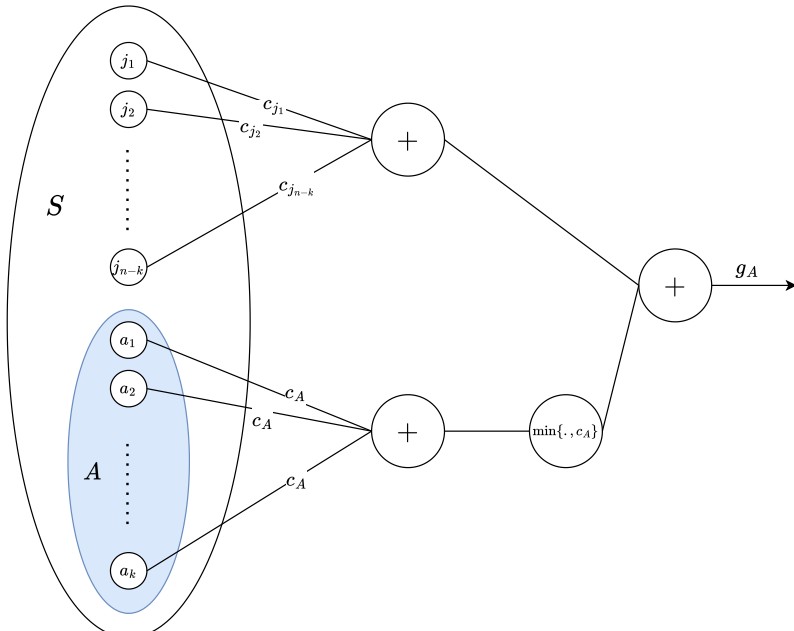

Figure 4: The simple architecture for the representation of $g_A$ as a DSF. The input of the network is a 0-1 vector corresponding to the subset $B$. The subset $B$ could have non-empty intersection with subset $A = \{a_1, \ldots, a_k\}$ and $S \setminus A = \{j_1, \ldots, j_{n-k}\}$.

**Lemma 3.5.** *For any $A \subseteq S$, we have*

$$\mathcal{P}_{g_A} = L_A. \tag{15}$$

*Proof.* To show the equality of two aforementioned sets, we first show $\mathcal{P}_{g_A} \subseteq L_A$, then we show $L_A \subseteq \mathcal{P}_{g_A}$, which completes the proof.

For the first part, for any $\mathbf{x} \in \mathcal{P}_{g_A}$, we have for all $B \subseteq S$, $\mathbf{x}(B) \leq g_A(B)$, therefore, for any $B \subseteq A$, we have $\mathbf{x}(B) \leq c_A$, in fact in this scenario $g_A(B) = 0$ or $c_A$, that implies $\mathbf{x}(A) \leq c_A$. Furthermore, for any $j \notin A$ we have $x_j \leq f(j) = c_j$. Therefore, each point in the polymatroid has the conditions to be in the $L_A$, that means, $\mathbf{x} \in L_A$.

For the reverse part, for any $\mathbf{x} \in L_A$, we have $\mathbf{x}(A) \leq c_A$, which implies that $\forall B \subseteq A, B \neq \emptyset$, $\mathbf{x}(B) \leq c_A = g_A(B)$. For any one-member subset of $S$, for example $B = \{j\}, j \notin A$ we have $x_j \leq c_j = g_A(B)$. Furthermore, for any arbitrary $B \subseteq S$, we could write $B = (A \cap B) \cup (B \setminus A)$. We have two cases,

1. $(A \cap B) = \emptyset$

$$\implies \mathbf{x}(B) = \sum_{j \in B \setminus A} x_j \leq \sum_{j \in B \setminus A} c_j = g_A(B)$$

2. $(A \cap B) \neq \emptyset$

$$\implies \mathbf{x}(B) \leq c_A + \sum_{j \in B \setminus A} x_j \leq c_A + \sum_{j \in B \setminus A} c_j = g_A(B).$$

It shows that for all $B \subseteq S$ we have: $\mathbf{x}(B) \leq g_A(B) \implies \mathbf{x} \in \mathcal{P}_{g_A}$. Therefore, the second part is now obvious, namely, $L_A \subseteq \mathcal{P}_{g_A}$. Hence,

$$\begin{cases} \mathcal{P}_{g_A} \subseteq L_A \\ L_A \subseteq \mathcal{P}_{g_A} \end{cases} \implies L_A = \mathcal{P}_{g_A}, \tag{16}$$

which completes the proof. □

For the next step, we introduce the next lemma.

**Lemma 3.6.** *For a given submodular function $f$, we have:*

$$\bigcap_{A \subseteq S} L_A = \mathcal{P}_f. \tag{17}$$

*Proof.* Similar to the proof of Lemma 3.5, we first show that $\bigcap_{A \subseteq S} L_A \subseteq \mathcal{P}_f$, then we will prove $\mathcal{P}_f \subseteq \bigcap_{A \subseteq S} L_A$.

For the first part, for any $\mathbf{x} \in \bigcap_{A \subseteq S} L_A$ we have

$$\forall A \subseteq S : \mathbf{x} \in L_A \implies \mathbf{x}(A) \leq c_A \implies \mathbf{x} \in \mathcal{P}_f. \tag{18}$$

Therefore, we have shown that $\bigcap_{A \subseteq S} L_A \subseteq \mathcal{P}_f$. For the reverse direction, for any $\mathbf{x} \in \mathcal{P}_f$, we can write

$$\forall A \subseteq S \implies \begin{cases} \mathbf{x}(A) \leq f(A) = c_A \\ \forall j \notin A : x_j \leq f(\{j\}) = c_j \end{cases} \implies \mathbf{x} \in L_A. \tag{19}$$

Therefore, we have shown that $\mathcal{P}_f \subseteq \bigcap_{A \subseteq S} L_A$. Combining these two parts completes our proof, namely, we have shown that $\bigcap_{A \subseteq S} L_A = \mathcal{P}_f$. □

For the next step, we need to present the following lemma.

**Lemma 3.7.** *For a given set $f_1, f_2, \ldots, f_r$ of submodular functions, if the function $h = \min(f_1, f_2, \ldots, f_r)$ is submodular, for the polymatroid of the function $h$, we have*

$$\mathcal{P}_h = \bigcap_{i=1,\ldots,r} \mathcal{P}_{f_i}. \tag{20}$$

*Proof.* To prove the lemma we proceed as follows.

$$\begin{aligned} \mathbf{x} \in \mathcal{P}_h &\iff \forall A \subseteq S : \mathbf{x}(A) \leq h(A) \\ &\iff \forall A \subseteq S : \forall i : \mathbf{x}(A) \leq f_i(A) \\ &\iff \forall i : \mathbf{x} \in \mathcal{P}_{f_i} \iff \mathbf{x} \in \bigcap_{i=1,\ldots,r} \mathcal{P}_{f_i}. \end{aligned}$$

□

In the next step, we introduce a new component to the DSF architecture to maintain the polymatroid corresponding to the output function to be the intersection of the polymatroids corresponding to the input functions. The proposed component is called *Min-Component*, as defined below.

**Definition 3.8.** (Min-Component) A Min-Component in the neural network, is a component (node) in which it brings the minimum of the inputs as the output of the node.

Incorporating this component into DSF, enhances its capability of representing submodular functions effectively.

In general, note that the min operator does not maintain the submodularity of the input functions. However, during the proof of the Theorem 3.9, we have used some techniques to assure that the output function would be submodular, as it can be seen in the following.

Moving towards the last stage, we present an architecture that is built upon the provided submodular function. In the initial layers, for any subset $A_i \subseteq S$, we have crafted the functions $g_{A_i}$ using the architecture outlined in Figure 4. Subsequently, we applied a min-component to all of these constructed $g_{A_i}$ functions, resulting in a composite function denoted as $g$. Therefore, the corresponding EDSF is generated.

In the following theorem[3], we establish that the function $g$ can precisely represents the original submodular function $f$.

**Theorem 3.9.** *Let $M = \{A_1, A_2, \ldots, A_{2^n}\} = 2^S$ represents all of the subsets of ground set $S$. The function $g = \min\{g_{A_1}, g_{A_2}, \ldots, g_{A_{2^n}}\}$ exactly represents the function $f$. In other words*

$$\forall A \subseteq S : g(A) = f(A). \tag{21}$$

*Proof.* Utilizing lemmas 3.5, 3.6, and 3.7, we can deduce that the polymatroid associated with the function $g$ is precisely identical to the polymatroid of the function $f$: $\mathcal{P}_g = \mathcal{P}_f$. However, it's worth noting that the function $g$ may belong to a broader category of functions than just submodular functions.

Based on lemma 2.9, we know that there exists some $\mathbf{x}^*$ such that $\mathbf{x}^* = \operatorname*{argmax}_{\mathbf{x} \in \mathcal{P}_f} \mathbf{x}(A)$ and $\mathbf{x}^*(A) = f(A)$.

Now suppose that $f(A) > g(A)$. We can write

$$\begin{cases} \mathbf{x}^* \in \mathcal{P} = \mathcal{P}_f = \mathcal{P}_g \\ \mathbf{x}^*(A) = f(A) \end{cases} \implies \mathbf{x}^*(A) = f(A) > g(A)$$

$$\implies \mathbf{x}^* \notin \mathcal{P}_g \implies \mathbf{x}^* \notin \mathcal{P}_f. \tag{22}$$

That is a contradiction, so for any set $A \subseteq S$ we have $g(A) \geq f(A)$.

Furthermore, we can write

$$\begin{cases} g(A) = \min_{B \subseteq S} \{g_B(A)\} \\ g_A(A) = c_A = f(A) \end{cases} \implies g(A) \leq f(A). \tag{23}$$

Combining these two results, it is clear that for any $A \subseteq S$ we have $g(A) = f(A)$. $\qquad \square$

Up to here, we have proved that all the monotone submodular functions can be represented by an EDSF function. In the remaining of this section, we will prove that this result also holds for the family of all monotone set functions.

**Lemma 3.10.** *For any EDSF function $f$, $f$ is monotone.*

*Proof.* The proof is straightforward, as all the weights used in construction of any EDSF is non-negative (all the constructing DSFs are monotone). All other functions that used in the construction of any EDSF such as min function are all monotone functions. $\qquad \square$

Next we prove the final result.

**Theorem 3.11.** *For any monotone set function $f$, there exists an EDSF $g$, such that $f = g$.*

*Proof.* Consider any monotone set function $f$. For any $B \subseteq S$ we define $g_B$ as follows

$$g_B(A) = \min \left\{ \sum_{j \in A \cap B} f(B), f(B) \right\} + \sum_{k \in A \setminus B} w^* \tag{24}$$

where $w^* = f(S)$, which is the maximum value function $f$ can take (because $f$ is monotone). Now we define function $g$ as follows:

$$g(A) = \min_{B \subseteq S} \{g_B(A)\}. \tag{25}$$

---

[3]Refer to Remark 5.2 for more discussion about Theorem 3.9. Moreover, in the review process, a simpler proof was also suggested by the anonymous reviewer, which for the sake of completeness have been presented in Appendix C.

For any $A \subseteq S$ we can see that $g_B(A) = f(A)$ if $B = A$. Now suppose $B \neq A$. If $A \setminus B \neq \emptyset$, $g_B(A) \geq f(S) \geq f(A)$, because of monotonicity. On the other hand if $A \setminus B = \emptyset$ we know that $A \subseteq B$. In this case we have $g_B(A) = f(B) \geq f(A)$ , because of monotonicity. Hence we can see that $g_A(A) \leq g_B(A)$ for any $B \subseteq S$. Therefore, $g(A) = f(A)$ for all $A \subseteq S$. $\qquad\square$

Based on the above results, we can conclude Theorem 3.2.

Although in our proofs we use exponential $r$ (number of DSFs), we observe in our experiments that in practice using much fewer DSFs is sufficient to learn coverage and monotone cut functions effectively.

## 3.2  Concavity of EDSFs

We can also show that any $g \in EDSFs$ is concave, if the input vector components are all non-negative real numbers, as stated in the following theorem.

**Theorem 3.12.** *Given $g \in EDSFs$, $g$ is a concave function with respect to the input vector, if the input vector components are all non-negative real numbers.*

*Proof.* Since all the DSF functions are concave in this setting Bai et al. (2018) (See Corollary 1), and $g$ is the minimum of a number of DSFs and we know that minimum of concave functions are concave, we can conclude that $g$ is also concave. $\qquad\square$

We can exploit the above-mentioned property to solve certain combinatorial optimization problems, such as the social welfare maximization problem, using gradient-based methods in an efficient manner. This provides a powerful tool to handle some combinatorial problems. Applications in this context are discussed in Section $4^4$.

# 4  Experimental Results

In the following section, we showcase a series of experiments aimed to demonstrating the positive outcomes and advantages derived from the application of Extended Deep Submodular Functions (EDSFs) in the modeling of submodular functions. Additionally, we highlight their efficacy in efficiently addressing and solving various combinatorial optimization problems. Through these experiments, we aim to provide a clear and comprehensive understanding of how EDSFs contribute to improved outcomes in the domain of submodular function modeling and the optimization of complex combinatorial scenarios.

## 4.1  Learning Coverage Functions

As outlined in Section 2, coverage functions constitute a crucial and intricate subset of monotone submodular functions, posing challenges in accurate learning from their instances.

Our experimental findings indicate that, in contrast to Deep Submodular Functions (DSFs), Extended Deep Submodular Functions (EDSFs) shown to be effective in efficiently learning these complex functions, exhibiting much lower empirical generalization error, compared to DSFs.

To perform our experiments, from each coverage function (defined in Definition 2.11), we generate a random dataset $\mathcal{D} = (X_i, y_i)_{i=1}^{d}$ where $X_i$ is a random subset from the ground set $S$, and $y_i$ is the value of the coverage function. To create a coverage function for our experiments, we define the universe size, the number of items (subsets), and the probability that each element in the universe independently belongs to each subset. Additionally, the weights in the coverage function are kept constant with a value of 1. For each experiment we generate a dataset, allocating 80% for training and 20% for testing.

The learning setup for all experiments in this section is the same. The optimizer used is Adam with a learning rate of 0.01. Each model is trained on a dataset of 1024 samples for 10,000 epochs. The employed

---

[4]Note that in all the problems described later in the paper, a set function is involved where the input is determined by a binary vector. Therefore, Theorem 3.12 can be applied.

cost function is the L1-loss function. Additionally, the weights for all EDSF and DSF neural networks are initialized using a Gaussian distribution with a mean of 0 and a variance of 0.01

In our first experiment, both an EDSF and DSF were trained to learn a coverage function with a universe size of 100, 16 subsets (items), and probability of 0.2. The architectures of the EDSF and DSF are quite similar, with the main difference being the min-component at the end of the EDSF. Each model consists of three fully-connected layers with 64 neurons, using $\phi(x) = \min(\alpha, x)$ (a minimum linear unit) as the activation function, where $\alpha = 95$. Additionally, we compared the performance of EDSF against other baselines, including the methods of Zaheer et al. (2017) and Lee et al. (2019), in learning the target coverage function. The neural networks were trained using the setup described above. Table 1 presents the train and test losses for each model, showing the mean and standard deviation across 20 runs. As shown, the EDSF demonstrates a significant improvement over the other baselines.

| Learning Coverage Function Experiment | | | | | | | |
|---|---|---|---|---|---|---|---|
| EDSF | | | | DSF | | | |
| Train | | Test | | Train | | Test | |
| mean | std | mean | std | mean | std | mean | std |
| 1.239 | 0.162 | 1.344 | 0.192 | 34.108 | 0.786 | 34.252 | 1.156 |
| Deep Sets | | | | Set Transformer | | | |
| Train | | Test | | Train | | Test | |
| mean | std | mean | std | mean | std | mean | std |
| 17.916 | 0.717 | 17.525 | 1.429 | 18.045 | 0.590 | 17.617 | 1.273 |

Table 1: Learning coverage function with universe size of 100, probability 0.2, and 16 items. The optimizer is Adam and learning rate is 0.01, using $L$1-Loss function to optimize. Experiments conducted on EDSF, DSF, Deep Sets, and Set Transformers. The value of $L$1-Loss function at the end of training and testing reported.

Additionally, the loss functions and outputs of the EDSFs and DSFs in learning coverage functions are shown in Appendix A, Figures 5-10. As observed, the DSFs were unsuccessful in these scenarios, with outputs remaining constant—likely due to the complexity of the coverage function. In contrast, the EDSFs demonstrated robust generalization, closely following the pattern of the target coverage function with low error, thus showcasing their superior performance in this experimental context.

Building on our first experiment, we tested various architectures for DSFs, and the training results were significantly worse than the above EDSF's results, as shown below:

1. 5 hidden layers with 64 neurons each, alpha set to 95: test loss is 84.3058

2. 4 hidden layers with 64 neurons each, alpha set to 125: test loss is 84.7727

3. 5 hidden layers with respectively 32, 64, 128, 64, and 32 neurons, alpha set to 400: test loss is 81.4188

These experimental results demonstrate the superiority of EDSF compared to DSF in learning the coverage functions. Additional DSF architectures (including larger ones) were also tested, and the corresponding plots are presented in Appendix A, Figures 11 and 12.

At the end of our first set of experiments, we conducted additional tests to examine the effects of various activation functions on the performance of DSFs during training. We tested three activation functions:

1. $a(x) = \log(1 + x)$

2. $a(x) = \tanh(x)$

3. $a(x) = \sigma(x) - 0.5$

The resulting plots for the loss functions, training, and testing performance are provided in Appendix A, Figures 13-15. As shown, none of these activation functions improved DSF performance in learning the

coverage function. Additionally, as discussed in Section 5, Remark 5.4, the only effective activation function for training EDSFs is MiLU, while none of these activation functions performed well when learning the coverage function with EDSFs.

In our second experiment, we tested different values of $r$, the number of DSFs used before the min function, to examine the effect of the EDSF network size on the generalization error of the training. The target function is a coverage function with a universe size of 500, 16 items, and a probability of 0.2. All experimental setups are the same as in our first experiment, with the only difference being the last layer, which now reflects the value of $r$, meaning that our EDSF has 4 layers with 64, 64, 64, and $r$ neurons, respectively. Table 2 presents the test loss for various values of $r$, showing the mean and standard deviation across 20 runs.

Further details about all the experiments can be found in Appendix A.

| r | Test Loss | |
|---|---|---|
| | mean | std |
| 1 | 88.587 | 7.308 |
| 2 | 86.664 | 4.614 |
| 4 | 88.499 | 5.915 |
| 8 | 87.759 | 5.305 |
| 16 | 87.236 | 6.02 |
| 32 | 3.693 | 0.539 |
| 64 | 4.332 | 0.695 |
| 128 | 4.633 | 1.136 |
| 256 | 4.899 | 2.091 |
| 512 | 7.674 | 3.708 |
| 1024 | 10.291 | 6.826 |

Table 2: Experiments for various size of EDSF in learning coverage functions. Here, $r$ is the number of DSF functions used in constructing the EDSF. The ground truth is coverage function with universe size 500, 16 items, and probability 0.2. The architecure of this experiment is same as Table 1, with only a difference in the last layer, showing value of $r$. The optimizer and learning rate used for each experience was, Adam and 0.01 respectively.

## 4.2 Learning Cut Functions

We conducted several experiments to learn modified graph cut functions (we choose modified version to maintain the monotonicity), which is a well-known submodular function. Firstly, we generated the random graph using Erdos-Renyi model (with probability 0.2 and 50 vertices), then, for any set of vertices $X$, we considered the function $f(X) = |\text{cut}(X)| + \sum_{a \in X} \deg(a)$ which is a monotone submodular function.

The architecture used to train the EDSF on the data is identical to the one used for learning coverage functions. It consists of three fully-connected layers with 64 neurons, each using $\phi(x) = \min(\alpha, x)$ (a minimum linear unit) as the activation function, where $\alpha = 95$. In the last layer, we incorporated a min-component with $r = 64$.

Similar to the coverage function experiment, we employed DSF, Deep Sets, and Set Transformer to learn the cut function and compare their performance with our proposed EDSF. The DSF used in this experiment is identical to that used for learning coverage functions, consisting of three layers with 64 neurons each, except for the alpha parameter, which is set to 450. For training our network, we used the Adam optimizer with a learning rate of 0.01, employed the $L1$-loss function as our cost function, and trained each network for 10,000 epochs. Additionally, the weights for the neural networks were initialized using a Gaussian distribution with a mean of 0 and a variance of 0.01. Table 3 presents the train and test losses for each model, showing the mean and standard deviation across 20 runs

Further figures and details about the experiments can be found in Appendix B.

| Learning Cut Function Experiment | | | | | | | |
| --- | --- | --- | --- | --- | --- | --- | --- |
| EDSF | | | | DSF | | | |
| Train | | Test | | Train | | Test | |
| mean | std | mean | std | mean | std | mean | std |
| 4.783 | 0.720 | 4.912 | 0.704 | 121.319 | 6.187 | 121.309 | 9.853 |
| Deep Sets | | | | Set Transformer | | | |
| Train | | Test | | Train | | Test | |
| mean | std | mean | std | mean | std | mean | std |
| 121.352 | 6.191 | 121.123 | 9.888 | 142.616 | 64.956 | 157.843 | 109.829 |

Table 3: Learning a cut function generated by the Erdos-Renyi model with probability 0.2 and having 50 vertices. The optimizer is Adam and learning rate is 0.01, using $L1$-Loss function to optimize. Experiments conducted on EDSF, DSF, Deep Sets, and Set Transformers. The value of $L1$-Loss function at the end of training and testing reported.

### 4.3 Social Welfare Maximization

In these experiments, we leveraged the previously discussed concave property inherent in EDSFs to address the widely recognized combinatorial optimization problem of maximizing social welfare, as delineated in Section 2. In this specific context, each of the valuation functions, denoted as $v_1, v_2, \ldots, v_n$, is assumed to be an EDSF or a DSF, acquired through learning from samples collected from users. Our optimization problem, expressed in Equation 9, can be reformulated as follows,

$$
\begin{aligned}
\max_{a} \quad & SW(a) \triangleq \sum_{i=1}^{n} v_i(a_i), \\
s.t. \quad & \sum_{i=1}^{n} a_{ij} = 1, \text{for all} \quad j \in S, \\
& a_{ij} \in \{0, 1\}.
\end{aligned}
\tag{26}
$$

where $a$ is a $n \times m$ matrix, such that $a_{ij} \in \{0, 1\}$ meaning if item $j$ assigned to user $i$ and $a_i$ represents the $i$'th row of $a$, which is the items that were assigned to user $i$.

To solve this optimization problem, first we relax it to the following form,

$$
\begin{aligned}
\max_{a} \quad & SW(a) = \sum_{i=1}^{n} v_i(a_i), \\
s.t. \quad & \sum_{i=1}^{n} a_{ij} = 1, \text{for all} \quad j \in S, \\
& a_{ij} \in [0, 1].
\end{aligned}
\tag{27}
$$

As we mentioned, each of the $v_i$'s are EDSF or DSF, so they are concave, hence, the relaxed problem is clearly a convex problem, e.g., maximizing a concave function with convex constraints, and can be solved using convex optimization techniques such as projected gradient ascent. The projection step consists of $m$ distinct projections on probability simplex for each item. The algorithm pseudo-code is shown in Algorithm 1.

In this set of experiments, we assume there are 3 users (bidders), each with a coverage function as their true submodular valuation function. Subsequently, our network is trained to learn each bidder's valuation function. Based on these learned networks, gradient ascent is then used to find a semi-optimal allocation that maximizes social welfare. Finally, the estimated social welfare is calculated based on this maximizing allocation with the true valuation functions.

The learning setup for all the experiments in this section is the same. The optimizer is Adam with a learning rate of 0.01. Each model is trained with a dataset of size 64 for 10,000 epochs. The employed cost function

---

**Algorithm 1** Gradient Ascent
___

    **Input:** valuation functions $v_1, v_2, \ldots, v_n$, set of items $S$ ($|S| = s$), learning rate $\eta$
    Initialize $a = (0)_{ij}$
    Project each column of $a$ on the probability simplex
    **repeat**
        Compute gradient of $SW$ function in the point $a = g \in \partial SW$.
        $a = a + \eta.g$
        Project each column of $a$ on the probability simplex
    **until** convergence
    **for** $i$ in $S$ **do**
        Select the user assigned to item $i$ by sampling from $i$'th column of $a$ (corresponding distribution for item $i$).
    **end for**
___

is the $L1$-loss function. Additionally, the weights for all neural networks are initialized with a Gaussian distribution with a mean of 0 and a variance of 0.01. Finally, for the gradient ascent part, the learning rate ($\eta$) is 0.001.

For our first experiment, we assume that the coverage function for each bidder has a universe size of 60 and 8 items. However, the coverage probabilities are 0.1, 0.3, and 0.5 for the three bidders, respectively. The learning model is an EDSF with 4 layers of 64 neurons, using MiLU with $\alpha = 95$ as its activation function. Three neural networks are trained, one for each bidder, following the learning setup described above. Table 4 shows the predicted social welfare and the optimal social welfare (calculated using brute force on the true valuation functions) for 10 different experiments.

| Exp. No. | Predicted Social Welfare | Optimal Social Welfare | Efficiency % |
|:---:|:---:|:---:|:---:|
| 1 | 124 | 127 | 97.6378 |
| 2 | 114 | 123 | 92.6829 |
| 3 | 84 | 129 | 65.1163 |
| 4 | 118 | 128 | 92.1875 |
| 5 | 114 | 127 | 89.7638 |
| 6 | 126 | 128 | 98.4375 |
| 7 | 110 | 122 | 90.1639 |
| 8 | 123 | 132 | 93.1818 |
| 9 | 123 | 128 | 96.0938 |
| 10 | 120 | 129 | 93.0233 |
| Average | 115.6 | 127.3 | 90.8288 |

Table 4: Experiments when value functions are coverage functions, with 60 universe size, and 8 items, with coverage probabilities ($\tilde{p}$) 0.1, 0.3, and 0.5 for three bidders respectively, and the learning model is EDSF. In the phase of learning EDSFs we have used the same setting as mentioned in the Table 1. The number of bidders ($n$) is 3 and the learning rate for the gradient ascent ($\eta$) is 0.001.

The second experiment is the same as the first experiment, except that our learning model is a DSF with 3 layers of 64 neurons, using the MiLU activation function with $\alpha = 95$. The corresponding results can be seen in Table 5. As observed, the average efficiency in this experiment is significantly lower than in the experiment with the EDSF model.

Furthermore, we conducted an experiment to compare the performance of vanilla neural networks versus EDSFs in estimating the optimal allocation for maximizing social welfare. By "vanilla neural network," we mean that weights can be negative, and activation functions may be non-concave. In this experiment, a

| Exp. No. | Predicted Social Welfare | Optimal Social Welfare | Efficiency % |
|---|---|---|---|
| 1 | 98 | 127 | 77.1654 |
| 2 | 98 | 121 | 80.9917 |
| 3 | 63 | 130 | 48.4615 |
| 4 | 50 | 124 | 40.3226 |
| 5 | 94 | 120 | 78.3333 |
| 6 | 92 | 131 | 70.2290 |
| 7 | 92 | 126 | 73.0159 |
| 8 | 99 | 129 | 76.7442 |
| 9 | 94 | 127 | 74.0157 |
| 10 | 60 | 125 | 48.0000 |
| Average | 84 | 126 | 66.7279 |

Table 5: Experiments when value functions are coverage functions, with 60 universe size, and 8 items, with coverage probabilities ($\tilde{p}$) 0.1, 0.3, and 0.5 for three bidders respectively, and the learning model is DSF. In the phase of learning DSFs we have used the same setting as mentioned in the Table 1. The number of bidders ($n$) is 3 and the learning rate for the gradient ascent ($\eta$) is 0.001.

fully-connected neural network with 3 layers, each having 64 neurons and using the ReLU activation function, was employed to learn the bidders' valuation functions. The networks were trained using the setup described above, and the results are shown in Table 6. We observe that the mean efficiency is about 77%, which is much lower than the EDSF efficiency of (90%).

| Exp. No. | Predicted Social Welfare | Optimal Social Welfare | Efficiency % |
|---|---|---|---|
| 1 | 56.0 | 98.0 | 57.14 |
| 2 | 79.0 | 101.0 | 78.28 |
| 3 | 89.0 | 93.0 | 95.70 |
| 4 | 81.0 | 94.0 | 86.17 |
| 5 | 70.0 | 99.0 | 70.71 |
| 6 | 52.0 | 94.0 | 55.32 |
| 7 | 84.0 | 100.0 | 84.00 |
| 8 | 62.0 | 101.0 | 61.39 |
| 9 | 88.0 | 99.0 | 88.89 |
| 10 | 81.0 | 96.0 | 84.37 |
| 11 | 77.0 | 93.0 | 82.80 |
| 12 | 92.0 | 108.0 | 85.19 |
| 13 | 77.0 | 104.0 | 74.04 |
| 14 | 85.0 | 99.0 | 85.86 |
| 15 | 69.0 | 94.0 | 73.40 |
| Average | 76.13 | 98.2 | 77.55 |

Table 6: Experiments when value functions are coverage functions, with 50 universe size, and 8 items, with probablities 0.1, 0.3, and 0.5 for three bidders respectively, and the learning model is vanilla neural network.The number of bidders ($n$) is 3 and the learning rate for the gradient ascent ($\eta$) is 0.001

Additionally, experiments with larger universe sizes of 500 and 1000 were conducted, and the corresponding results, comparing the performance of EDSF and DSF in predicting optimal social welfare, are shown in Tables 7 and 8, respectively. Other settings for these experiments are the same as those in Tables 4 and 5.

| Exp. No. | EDSF Predicted Social Welfare | DSF Predicted Social Welfare | Optimal Social Welfare | EDSF Eff. % | DSF Eff. % |
|---|---|---|---|---|---|
| 1 | 892.0 | 653.0 | 902.0 | 98.89 | 72.39 |
| 2 | 907.0 | 634.0 | 917.0 | 98.91 | 69.14 |
| 3 | 887.0 | 636.0 | 892.0 | 99.44 | 71.30 |
| 4 | 882.0 | 500.0 | 900.0 | 98.00 | 55.56 |
| 5 | 909.0 | 670.0 | 913.0 | 99.56 | 73.38 |
| Average | 894.0 | 618.0 | 904.0 | 98.96 | 68.35 |

Table 7: Experiments comparing EDSF and DSF efficiency in the maximizing social welfare problem, with coverage function as value function, 500 universe size, with probabilities 0.1, 0.3, and 0.5 for three bidders, respectively. In the phase of learning EDSFs and DSF we have used the same setting as mentioned in the Table 1. The number of bidders ($n$) is 3 and the learning rate for the gradient ascent ($\eta$) is 0.001

| Exp. No. | EDSF Predicted Social Welfare | DSF Predicted Social Welfare | Optimal Social Welfare | EDSF Eff. % | DSF Eff. % |
|---|---|---|---|---|---|
| 1 | 1737.0 | 1527.0 | 1782.0 | 97.47 | 85.69 |
| 2 | 1742.0 | 1730.0 | 1789.0 | 97.37 | 96.70 |
| 3 | 1744.0 | 1547.0 | 1778.0 | 98.09 | 87.01 |
| 4 | 1784.0 | 1739.0 | 1790.0 | 99.66 | 97.15 |
| 5 | 1755.0 | 1699.0 | 1776.0 | 98.82 | 95.66 |
| Average | 1752.4 | 1648.4 | 1783.0 | 98.2 | 92.60 |

Table 8: Experiments comparing EDSF and DSF efficiency in the maximizing social welfare problem, with coverage function as value function, 1000 universe size, with probabilities 0.1, 0.3, and 0.5 for three bidders, respectively. In the phase of learning EDSFs and DSF we have used the same setting as mentioned in the Table 1. The number of bidders ($n$) is 3 and the learning rate for the gradient ascent ($\eta$) is 0.001.

Finally, we conducted a set of experiments to demonstrate the optimal learned social welfare based on the trained EDSFs and DSFs. All details of this experiment are the same as in Tables 4 and 5, except for the universe size, which is set to 1000. Results are presented in Table 9. As observed, the optimal learned social welfare for EDSFs is close to the optimal social welfare, indicating a good estimation of the true valuation functions. However, the optimal learned social welfare for DSFs is significantly lower than the optimal value. Additionally, it is worth noting that the gradient ascent algorithm, when using EDSF networks, achieves reasonably good performance and shows only a slight difference from the optimal learned social welfare for EDSFs.

Observing the results, it becomes apparent that the utilization of Extended Deep Submodular Functions (EDSFs) in the context of the social welfare maximization problem yields significantly higher efficiency when compared to Deep Submodular Functions (DSFs) and also vanilla neural networks. This stark difference in efficiency underscores the potential advantages and superior performance that our proposed framework, leveraging EDSFs, can bring to the modeling of user valuations within the realm of this NP-hard combinatorial optimization problem.

## 5 Discussion and Remarks

In this paper, we introduced an architecture to represent all monotone set/submodular functions using neural networks. Here are some points about the proposed architecture that are worth-noting.

*Remark* 5.1 (Computational Efficiency). The most important shortcoming of this architecture is the exponential size of the network to represent all monotone set/submodular functions. However, we would like to note

| Exp. No. | EDSF Predicted SW | DSF Predicted SW | Optimal Learned EDSF SW | Optimal Learned DSF SW | Optimal SW | EDSF Eff. % | DSF Eff. % |
|---|---|---|---|---|---|---|---|
| 1 | 2340.0 | 1965.0 | 2390.0 | 2159.0 | 2390.0 | 97.91 | 82.22 |
| 2 | 2274.0 | 2024.0 | 2373.0 | 2016.0 | 2373.0 | 95.83 | 85.29 |
| 3 | 2348.0 | 2239.0 | 2382.0 | 2059.0 | 2387.0 | 98.37 | 93.8 |
| 4 | 2306.0 | 2187.0 | 2392.0 | 2037.0 | 2392.0 | 96.4 | 91.43 |
| 5 | 2300.0 | 1464.0 | 2366.0 | 2130.0 | 2372.0 | 96.96 | 61.72 |
| 6 | 2369.0 | 2268.0 | 2372.0 | 2051.0 | 2383.0 | 99.41 | 95.17 |
| 7 | 2262.0 | 1771.0 | 2375.0 | 2050.0 | 2387.0 | 94.76 | 74.19 |
| 8 | 2306.0 | 1977.0 | 2344.0 | 2309.0 | 2363.0 | 97.59 | 83.66 |
| 9 | 2360.0 | 2087.0 | 2360.0 | 2024.0 | 2378.0 | 99.24 | 87.76 |
| 10 | 2321.0 | 2075.0 | 2370.0 | 2008.0 | 2383.0 | 97.4 | 87.08 |
| Avg | 2318.6 | 2005.7 | 2372.4 | 2084.3 | 2380.8 | 97.7 | 84.5 |

Table 9: Experiments comparing EDSF and DSF efficiency in the maximizing social welfare problem, with coverage function as value function, 1000 universe size, with probabilities 0.1, 0.3, and 0.5 for three bidders, respectively. In the phase of learning EDSFs and DSF we have used the same setting as mentioned in the Table 1. The number of bidders ($n$) is 3 and the learning rate for the gradient ascent ($\eta$) is 0.001.

that the assumption on the exponentiality of the number of DSFs is just used as a sufficient condition for our proofs. In our experiments, esspecialy in Table 2 with different numbers of DSFs tested, we observed that using much fewer number of DSFs than exponential order is enough to attain good generalization in practice. For example, in the problem of learning coverage functions, we used 64 DSFs to represent a function with 16 and 50 items as input (note that $64 \ll 2^{16}$ and $64 \ll 2^{50}$). We have also used 64 DSFs in learning cut functions of graphs with 50 vertices ($64 \ll 2^{50}$).

*Remark* 5.2. Regarding Theorem 3.9, we would like to note that the architecture of EDSFs was derived through an analysis of polymatroids, which also forms the basis of proving Theorem 3.9. We believe that including Theorem 3.9, along with its proof based on polymatroid theory, is essential for a deeper understanding of EDSFs. Additionally, during the review process, a simpler proof of Theorem 3.9 was suggested by one of the reviewers, which we have included in Appendix C for completeness.

*Remark* 5.3. There exists some approximation algorithm to find the near-optimal social welfare in the problem setting mentioned above, like the one proposed in Vondrak (2008), namely, Continuous Greedy. We have conducted multiple experiments in order to compare the Continuous Greedy and Gradient Ascent algorithm as shown in Table 10.

As can be seen from the results, the "average" performance of both methods (Gradient Ascent and Continuous Greedy) is very similar, leading us to conclude that there may be a shared intuition underlying both algorithms. The intuition behind the continuous greedy algorithm is that the current distribution will shift towards the most increasing direction to find the optimal distribution for sampling. This concept is reminiscent of moving in the direction of the gradient in concave functions to maximize them. Overall, it appears that the Gradient Ascent algorithm closely resembles the continuous dual of the continuous greedy algorithm. Exploring the theoretical connection between Gradient Ascent and Continuous Greedy can be an interesting direction for future work.

However, in the following, we would like to mention that the continuous greedy algorithm has a major limitation.

As it can be seen in the Table 10, the running time of the Continuous Greedy algorithm is much larger than the Gradient Ascent in practice. It seems the reason behind this large difference is that the CG algorithm requires to sample from the function in order of $O((mn)^5)$ times to estimate the expectations in the algorithm. It makes it so computationally complex in practice. However, since the size of the neural network for the proposed method in practice is much smaller than the theoretical requirements, using GA is much faster than the CG algorithm, as it can be seen in the Table 10.

| Universe Size | Continuous Greedy Social Welfare | | Gradient Ascent Social Welfare | | Continuous Greedy Wall-Clock Time (s) | | Gradeint Ascent Wall-Clock Time(s) | |
|---|---|---|---|---|---|---|---|---|
| | mean | std | mean | std | mean | std | mean | std |
| 200 | 599.666 | 0.471 | 598. | 1.632 | 585.943 | 1.857 | 99.404 | 0.076 |
| 500 | 1499. | 0.816 | 1500. | 0.000 | 601.558 | 0.425 | 101.290 | 0.060 |
| 1000 | 2998. | 1.788 | 2997.8 | 1.326 | 625.598 | 3.089 | 103.320 | 0.446 |

Table 10: Experiments comparing Continuous Greedy and Gradient Ascent algorithms to maximize social welfare after training EDSFs on coverage function with different universe sizes. It shows that the CG algorithm in practice is much slower than the GA algorithm.

It is also worth noting that, there are some algorithms based on continuous greedy idea to find the maximizing input for submodular functions under different kinds of constraints Mokhtari et al. (2020); Badanidiyuru & Vondrák (2014) which are much faster than the original version of the algorithm in Vondrak (2008). But note that while these methods are faster than continuous greedy, their approximation factor is bounded by $1 - \frac{1}{e}$. However, we know that the approximation factor for maximizing monotone DSFs with a matroid constraint of rank $k$ using gradient ascent is $\max_{0 < \delta < 1} \left( 1 - \epsilon - \delta - e^{-\delta^2 \Omega(k)} \right)$ Bai et al. (2018), which is significantly better than $1 - \frac{1}{e}$ for large $k$. Note that the proof of the approximation factor for maximizing DSFs using gradient ascent (including pipage rounding as the last step) is exactly applicable to EDSFs. The intuition is that since for each input $x \in [0,1]^n$, only one of the DSFs is active, then for functions that can be represented by an EDSF using the minimum over a finite number of DSFs, we can achieve a better approximation factor than $1 - \frac{1}{e}$. Note that this is only interesting for EDSFs that we can represent using minimum over polynomial number of DSFs.

*Remark* 5.4. As shown in Appendix A, 5-7, the loss function for EDSF exhibits significant fluctuations and spikes when learning coverage functions. We hypothesized that this was due to the MiLU activation function, which has sharp edges that could complicate backpropagation and hinder the learning process. To test this conjecture, we conducted additional experiments with EDSF using different, smoother activation functions, such as $\log(1 + x)$. As seen in Figures 16-18, these loss functions became much smoother, with no visible fluctuations. However, these smoother activation functions did not generalize as effectively as MiLU and were unable to learn the coverage function as well.

*Remark* 5.5. Finally, we conclude our discussion by analyzing why DSFs produce constant outputs after training, particularly when learning coverage functions, as shown in Figures 9-10. To test the expressive power of DSFs, especially for coverage functions, we ran additional experiments with much larger DSFs. The results, presented in Figures 11-12, indicate that DSFs still output constant values. We hypothesize that this issue arises when neurons in a layer reach the saturation point of their activation function—a limitation observed across common concave activation functions. Beyond this saturation point, DSFs produce a constant output. Using larger DSFs exacerbates this issue, while smaller DSFs, as seen in Figure 8, can sometimes avoid it.

# 6 Conclusions

In this research, we introduce a novel concept called Extended Deep Submodular Functions (EDSFs), building upon the foundation of Deep Submodular Functions (DSFs). DSFs, a subset of monotone submodular functions, provide a structured framework for representing specific types (a subset) of submodular functions. However, the scope of DSFs is limited to a strict subset of monotone submodular functions. EDSFs, on the other hand, serve as a natural extension, expanding the family of DSFs to encompass all monotone set/submodular functions. Our proofs are rooted in the properties of polymatroids, offering insights into the relationship between polymatroids and submodular functions. Additionally, we highlight the concave nature of EDSFs, a characteristic that is proved to be valuable in addressing and efficiently solving various combinatorial optimization problems. To validate the efficacy of EDSFs, we conducted experiments in three distinct settings, namely, learning coverage functions, learning modified cut functions, and maximizing social

welfare. The results consistently demonstrated the superior performance of EDSFs compared to DSFs in these experimental scenarios. In conclusion, our findings suggest that EDSFs provide a more comprehensive and effective solution for modeling monotone set/submodular functions using neural networks. The extended scope and enhanced performance make EDSFs a promising direction for further exploration in various machine learning and optimization domains.

## Broader Impact

This paper presents work whose goal is to advance the field of Machine Learning. There aren't many straight potential societal consequences of our work, none which we feel must be specifically highlighted here.

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

## A   Details of Experiments on Learning Coverage Functions

To set the hyperparameters—such as the number of layers, neurons, and the value of alpha—we performed an exhaustive search across a defined parameter space. The search space included varying the number of layers from 2 to 6, the number of neurons per layer from 32 to 2048 (in powers of 2), and testing different values of alpha, starting from small values around 5 up to larger values in the order of 1000. While many different hyperparameter configurations worked well for the EDSF architecture, none of the candidates yielded promising results for DSFs in terms of output or loss. Additionally, for all weight initialization in our DSF

and EDSF networks across all experiments in the paper, we used a Gaussian distribution with a mean of 0 and a variance of 0.1.

For our first experiment (see Table 1), the architecture for the Deep Set model consists of three layers with 64 neurons each for $\phi$ and three layers with 64 neurons each for $\rho$. The architecture used for the Set Transformer model includes 4 attention heads and a hidden dimension of 128.

To demonstrate the loss function and outputs of EDSFs and DSFs in learning coverage functions, we tested three different coverage functions with probabilities of 0.1, 0.3, and 0.5. All three coverage functions had a universe size of 100 and 16 items. The DSF and EDSF used for these experiments had the same architecture as described in Table 1. For each experiment, our network was trained for 10,000 epochs using the Adam optimizer with a learning rate of 0.01. The employed cost function, consistent with all other experiments, was the $L1$-loss function. As shown in Figures 5-10, the DSF fails to learn the patterns of the target coverage functions and outputs a constant value. In contrast, the EDSF can closely follow the patterns of the target functions.

To demonstrate the expressive power of DSFs, we employed larger architectures for DSFs in learning the coverage function. The DSFs used in these experiments had 4 and 6 layers, each with 2048 neurons, and utilized the MiLU activation function with $\alpha = 95$. The target function was a coverage function with a universe size of 100, 16 items, and a probability of 0.5. Consistent with all other experiments, the network was trained for 10,000 epochs using the Adam optimizer with a learning rate of 0.01 and the $L1$-loss function as the cost function. As shown in Figures 11 and 12, the DSF still failed to learn the target coverage function, producing a constant output instead. Additional experiments using various concave activation functions for DSFs were also conducted to learn the aforementioned coverage function, with results presented in Figures 13-15. The learning setup for these experiments was consistent with the other experiments. Moreover, the DSF used for these experiments had 3 layers with 64 neurons each.

Finally, to test our conjecture described in Remark 5.4 regarding the loss function for EDSFs when learning coverage functions with many spikes and fluctuations, we experimented with smoother concave activation functions. As shown in the corresponding plots in Figures 16-18, the loss functions exhibit no visible spikes or fluctuations. For these experiments, the target coverage function had a universe size of 100, 31 items, and probability of 0.2. Moreover, The EDSF used for these experiments had 4 layers with 64 neurons each, similar to Table 1, and learning setup was consistent with other experiments.

## B   Details of Experiments on Learning Cut Functions

For setting hyperparameters in learning the cut function, similar to learning the coverage function, we conducted an exhaustive search and tried different values for the number of layers, neurons, and values of alpha. The search space for these parameters was the same as that used in learning coverage functions, as mentioned in Appendix A. Many different candidates for the DSF encountered the same problem of outputting a constant value; however, a few of them could approximately identify the pattern of the target cut function. It is worth mentioning that they could not learn the cut function as well as the EDSF, and most of the hyperparameters worked well for the EDSF.

For our first experiment in Section 4.2 (see Table 3), the architecture for the Deep Set model consists of three layers with 64 neurons each for $\phi$ and three layers with 64 neurons each for $\rho$. The architecture used for the Set Transformer model includes 4 attention heads and a hidden dimension of 128.

We can observe the loss function and the true vs. predicted values for the train and test samples for the EDSF and DSF in Figures 19 and 20, respectively. The setup for these experiments is the same as the setup described in Section 4.2, Table 3.

## C   Alternative proof for the main result of the paper

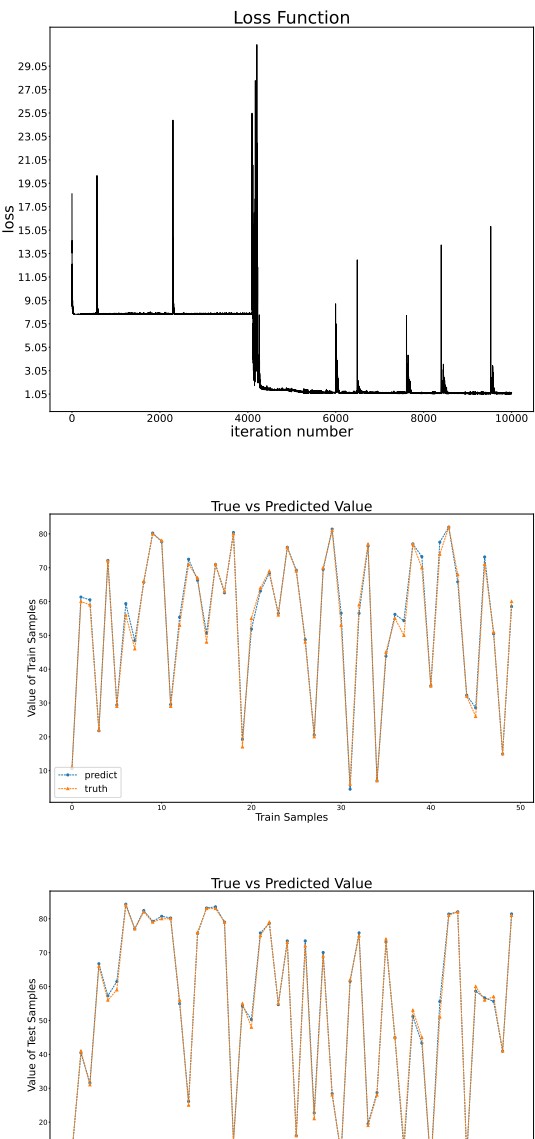

Figure 5: Learning coverage function with probability 0.1, universe size 100, and 16 items. We have used EDSF and shown Training loss, Truth vs. Predicted values for train and test samples. Other settings are same as Section 4.1 Table 1.

There is another much simpler proof to establish the main result of the paper without using polymatroid analysis. [5] If we define the function $g_A(B)$ as:

$$g_A(B) = \begin{cases} \sum_{j \in B} f(j) & \text{if } A \cap B = \emptyset \\ f(A) + \sum_{j \in B} f(j) & \text{if } A \cap B \neq \emptyset \end{cases},$$ (28)

hence, we can conclude that $f(B) \leq \min_A g_A(B)$ Fisher et al. (1978). Then, since we have $g_B(B) = f(B)$, we conclude that $f(B) = \min_A g_A(B)$ for all $B \subseteq S$. This proof is much simpler than the provided proof in

---

[5]This proof provided kindly by one of the reviewers during the review process.

the paper, however, without using the analysis of polymatroids there is no intuition at hand about how we constructed the architecture of the EDSFs to respresent all of the monotone submodular functions, which makes is possible to generalize the architecture to all monotone set functions as a more powerful result.

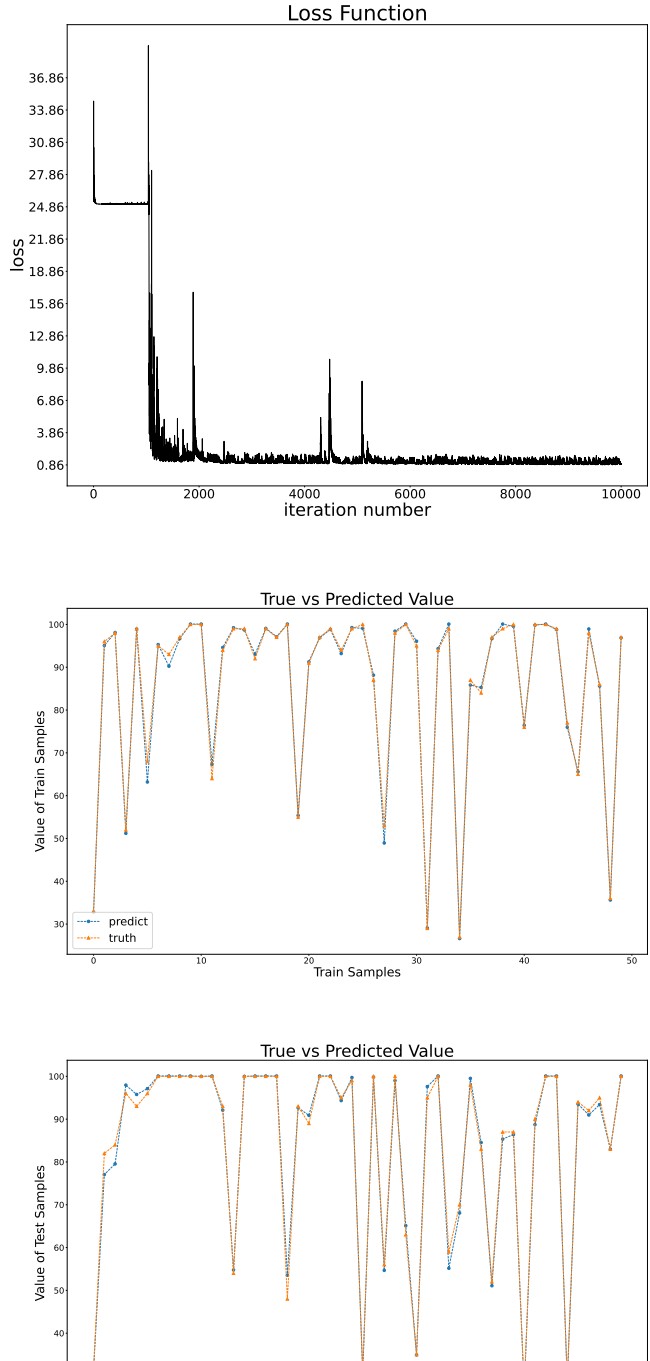

Figure 6: Learning coverage function with probability 0.3, universe size 100, and 16 items. We have used EDSF and shown Training loss, Truth vs. Predicted values for train and test samples. Other settings are same as Section 4.1 Table 1.

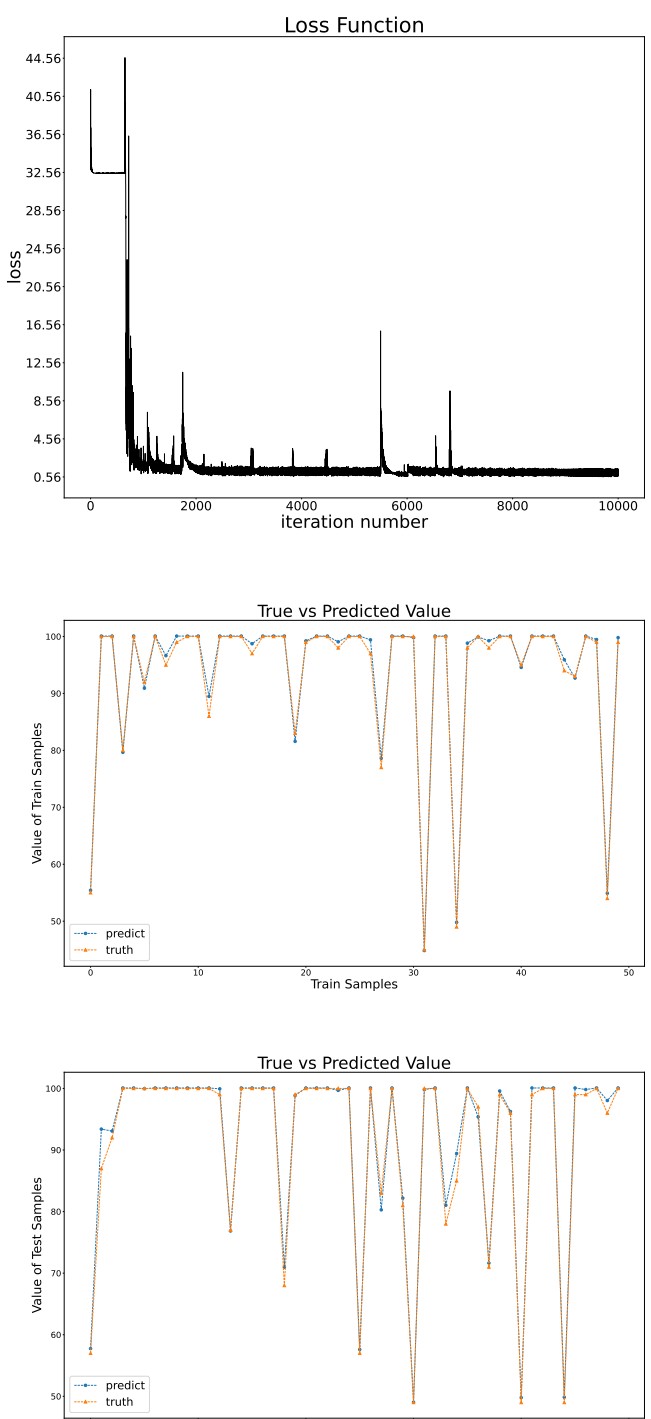

Figure 7: Learning coverage function with probability 0.5, universe size 100, and 16 items. We have used EDSF and shown Training loss, Truth vs. Predicted values for train and test samples. Other settings are same as Section 4.1 Table 1.

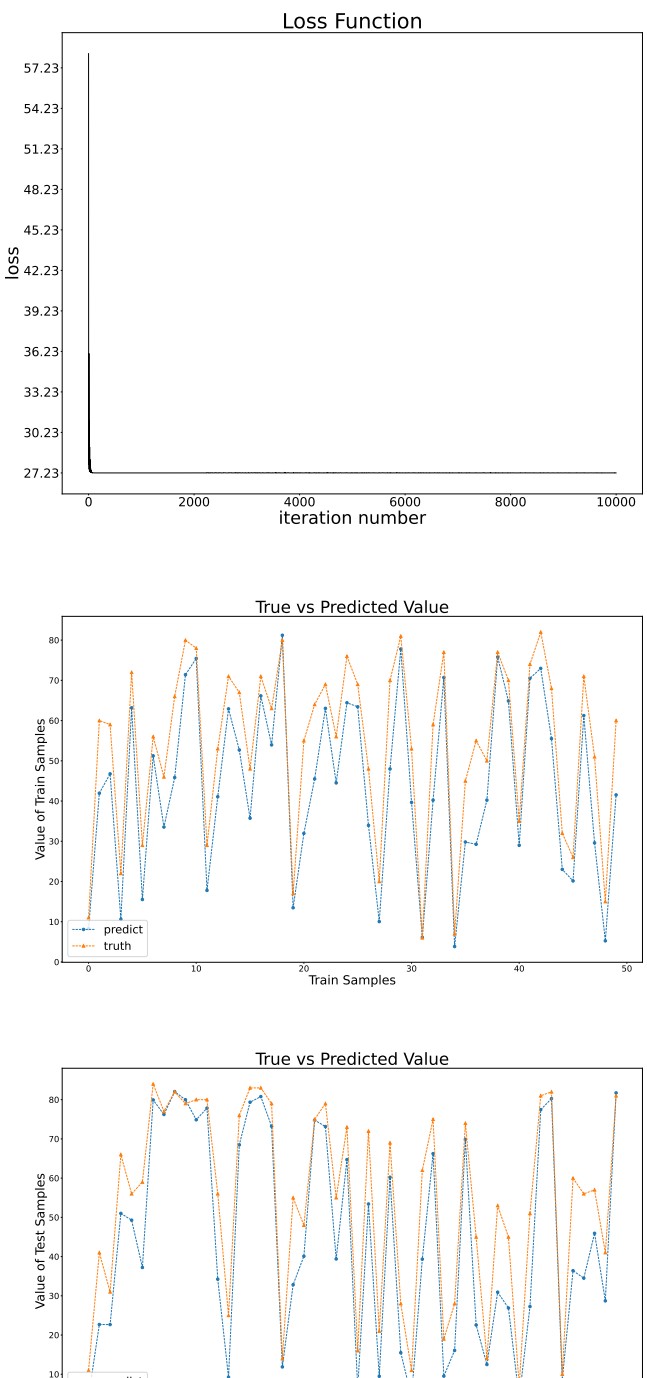

Figure 8: Learning coverage function with probability 0.1, universe size 100, and 16 items. We have used DSF and shown Training loss, Truth vs. Predicted values for train and test samples. Other settings are same as Section 4.1 Table 1.

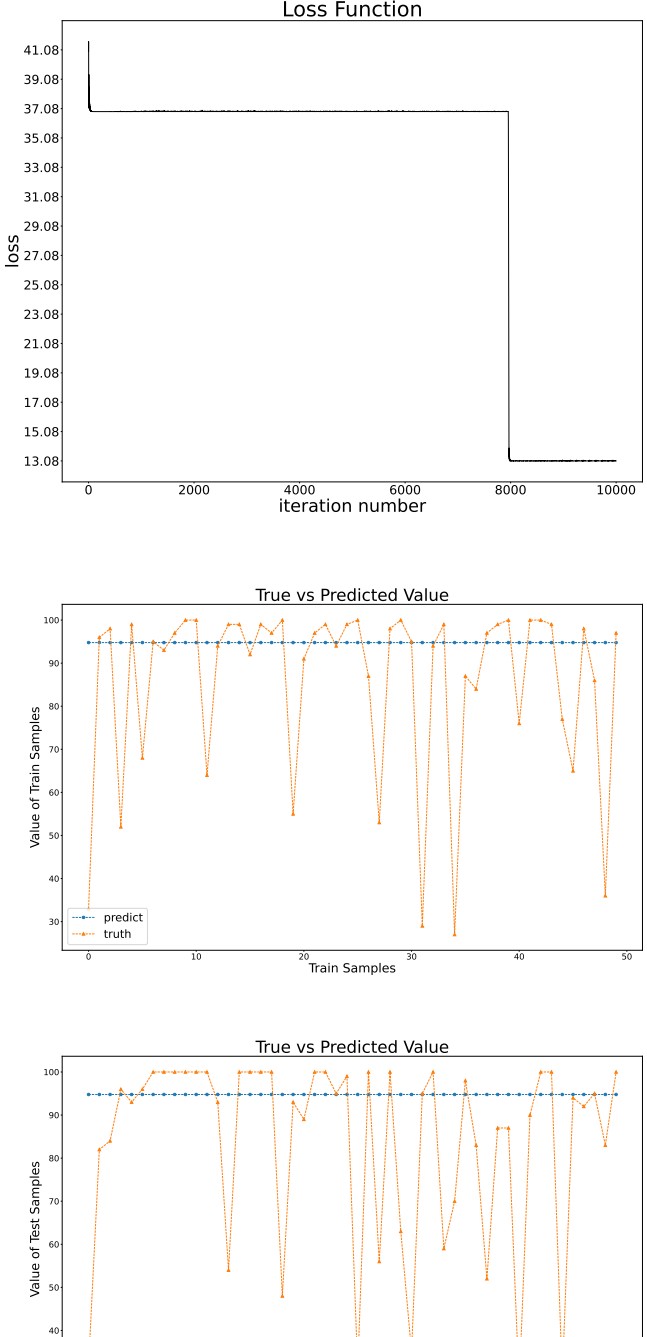

Figure 9: Learning coverage function with probability 0.3, universe size 100, and 16 items. We have used DSF and shown Training loss, Truth vs. Predicted values for train and test samples. Other settings are same as Section 4.1 Table 1.

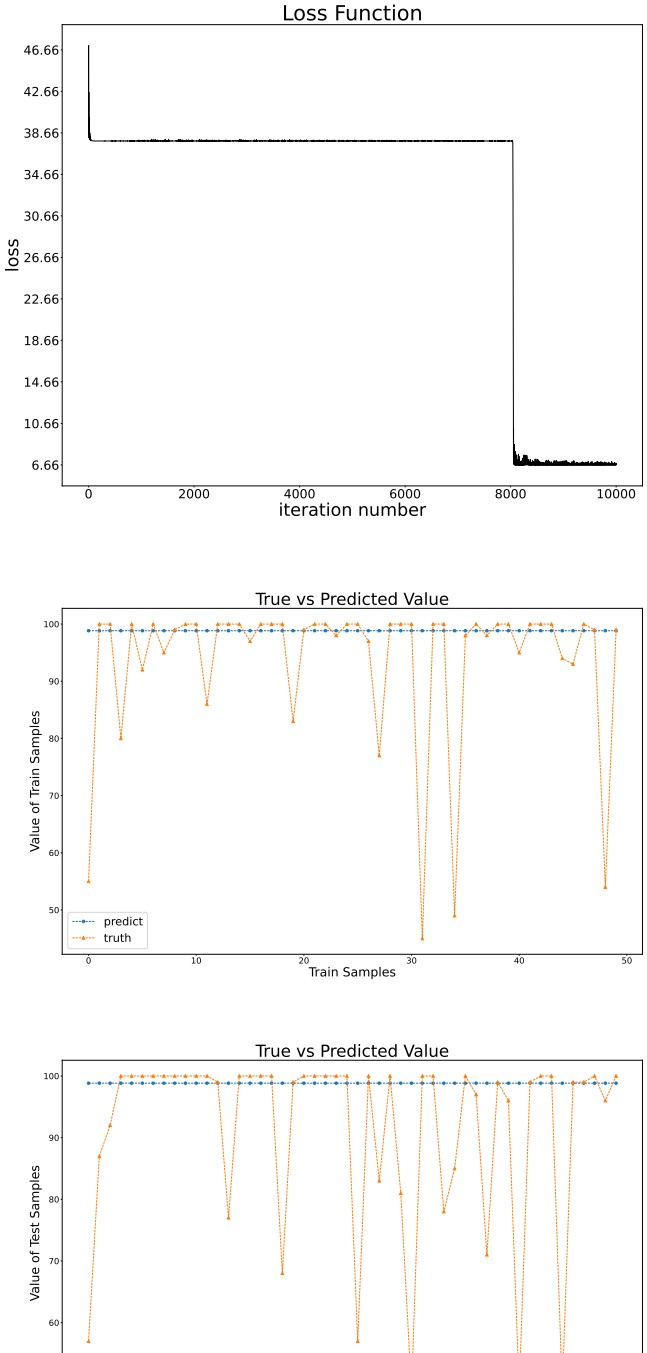

Figure 10: Learning coverage function with probability 0.5, universe size 100, and 16 items. We have used DSF and shown Training loss, Truth vs. Predicted values for train and test samples. Other settings are same as Section 4.1 Table 1.

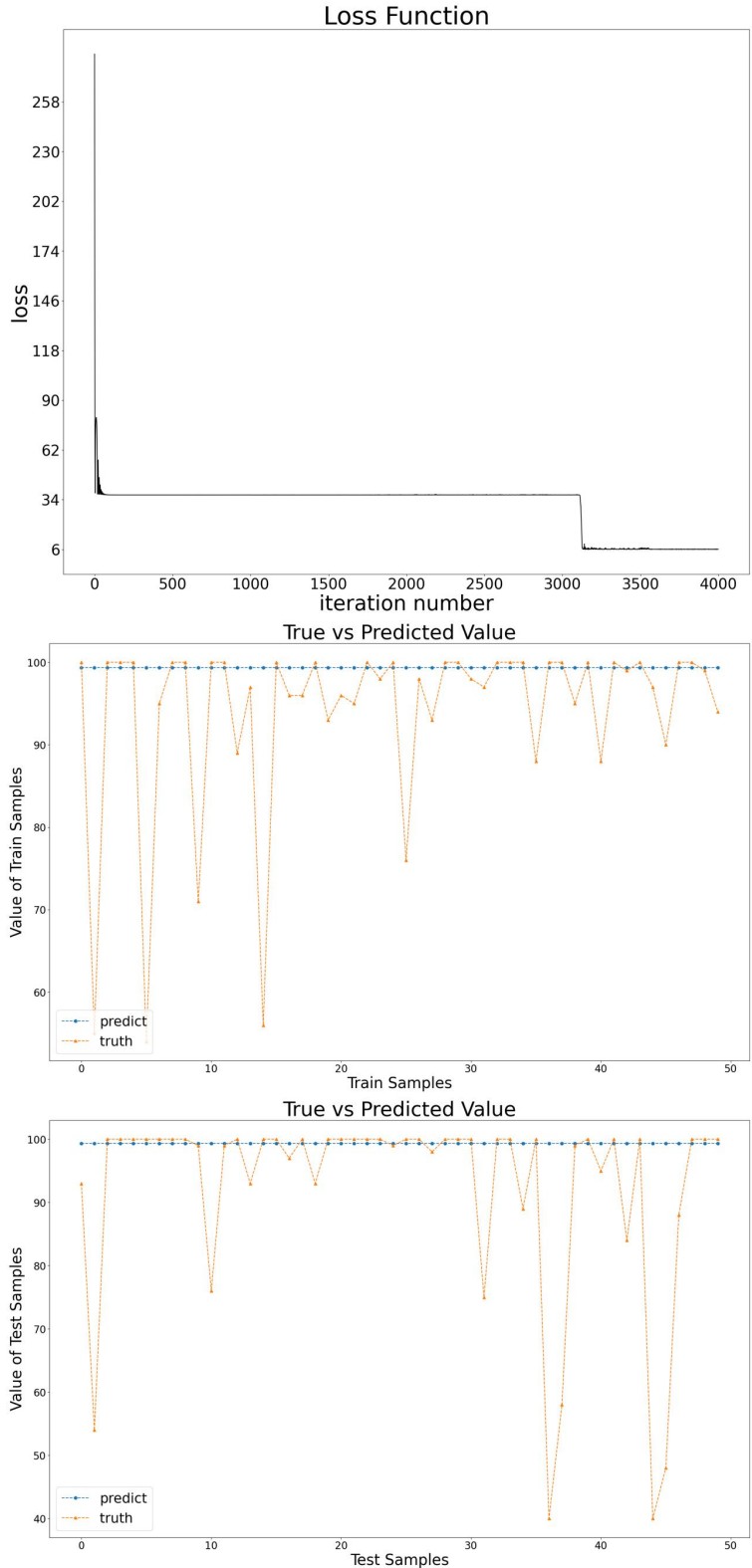

Figure 11: Learning coverage function with DSF having more number of neurons. The used DSF has 4 layers each having 2048 neurons and MiLU activation function with $\alpha = 95$. we can see that it still outputs constant when learning coverage function with universe size of 100 and probability of 0.5.

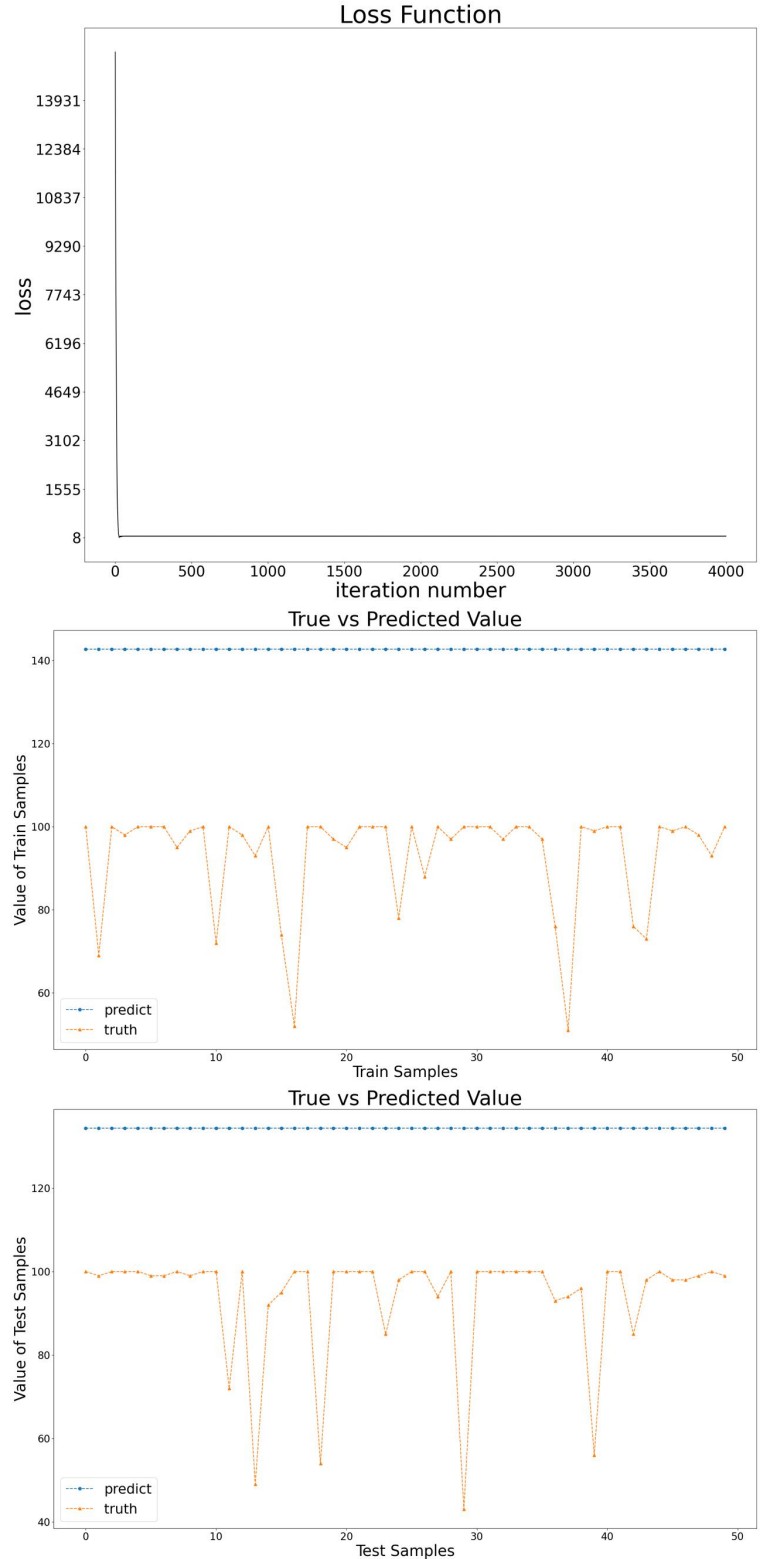

Figure 12: Learning coverage function with DSF having more number of neurons. The used DSF has 6 layers each having 2048 neurons and MiLU activation function with $\alpha = 95$. we can see that it still outputs constant when learning coverage function with universe size of 100 and probability of 0.5.

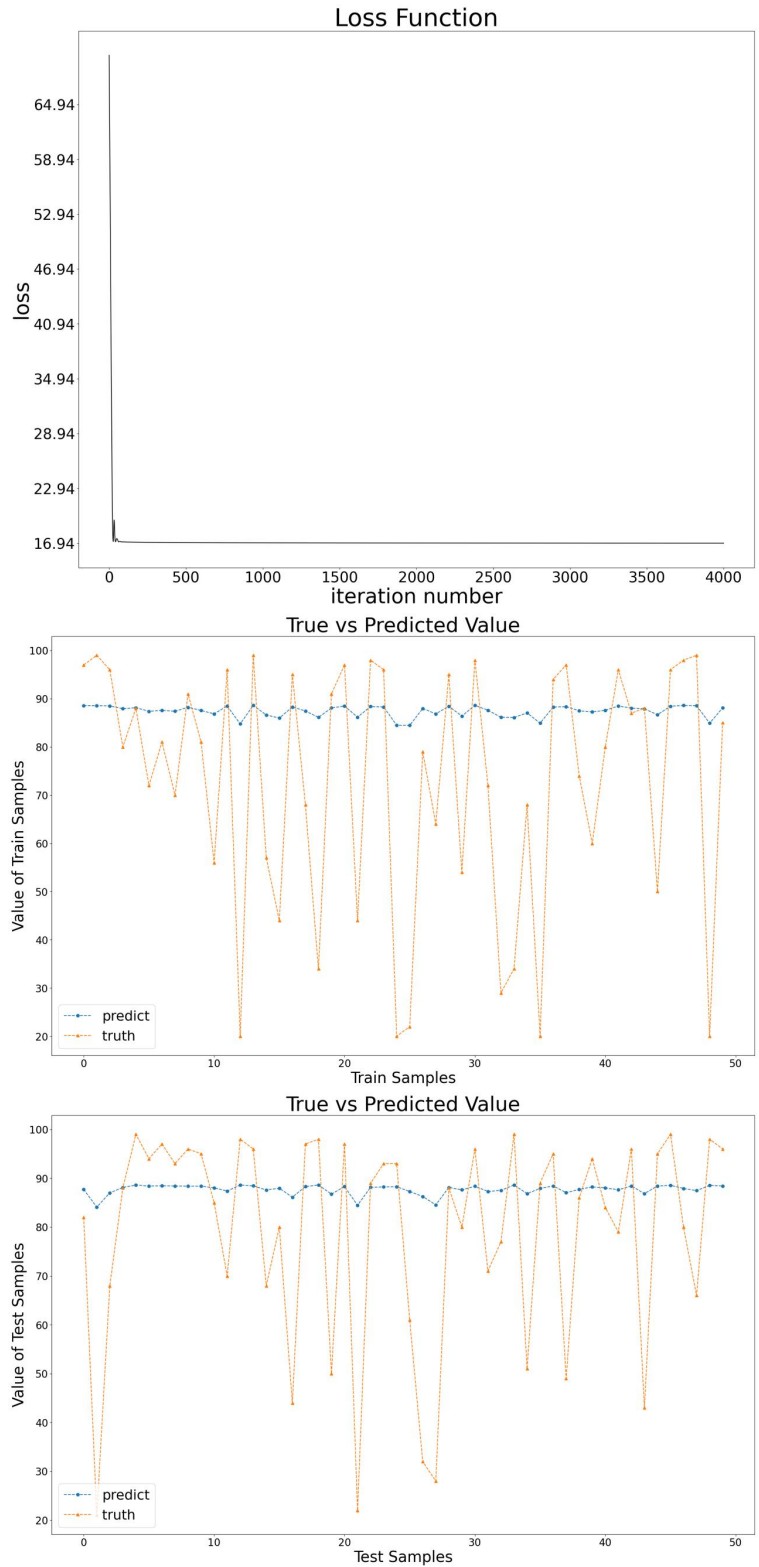

Figure 13: Learning coverage function with DSF having $\log(1 + x)$ as activation function. The used DSF has 3 layers each having 64 neurons and $\log(1 + x)$ as activation function. We can see that it still outputs constant when learning coverage function with universe size of 100 and probability of 0.5.

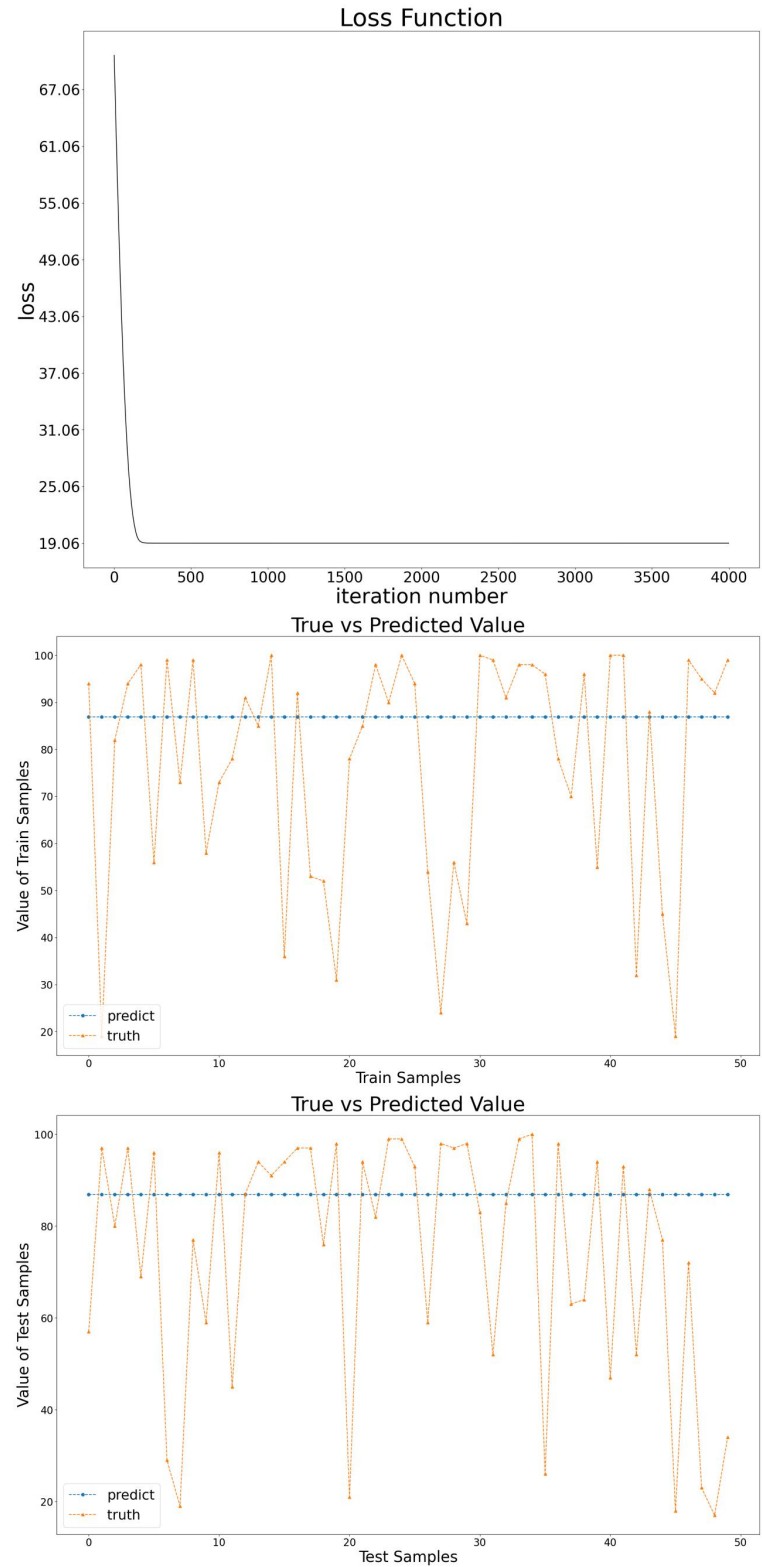

Figure 14: Learning coverage function with DSF having tanh($x$) as activation function. The used DSF has 3 layers each having 64 neurons and tanh($x$) as activation function. We can see that it still outputs constant when learning coverage function with universe size of 100 and probability of 0.5.

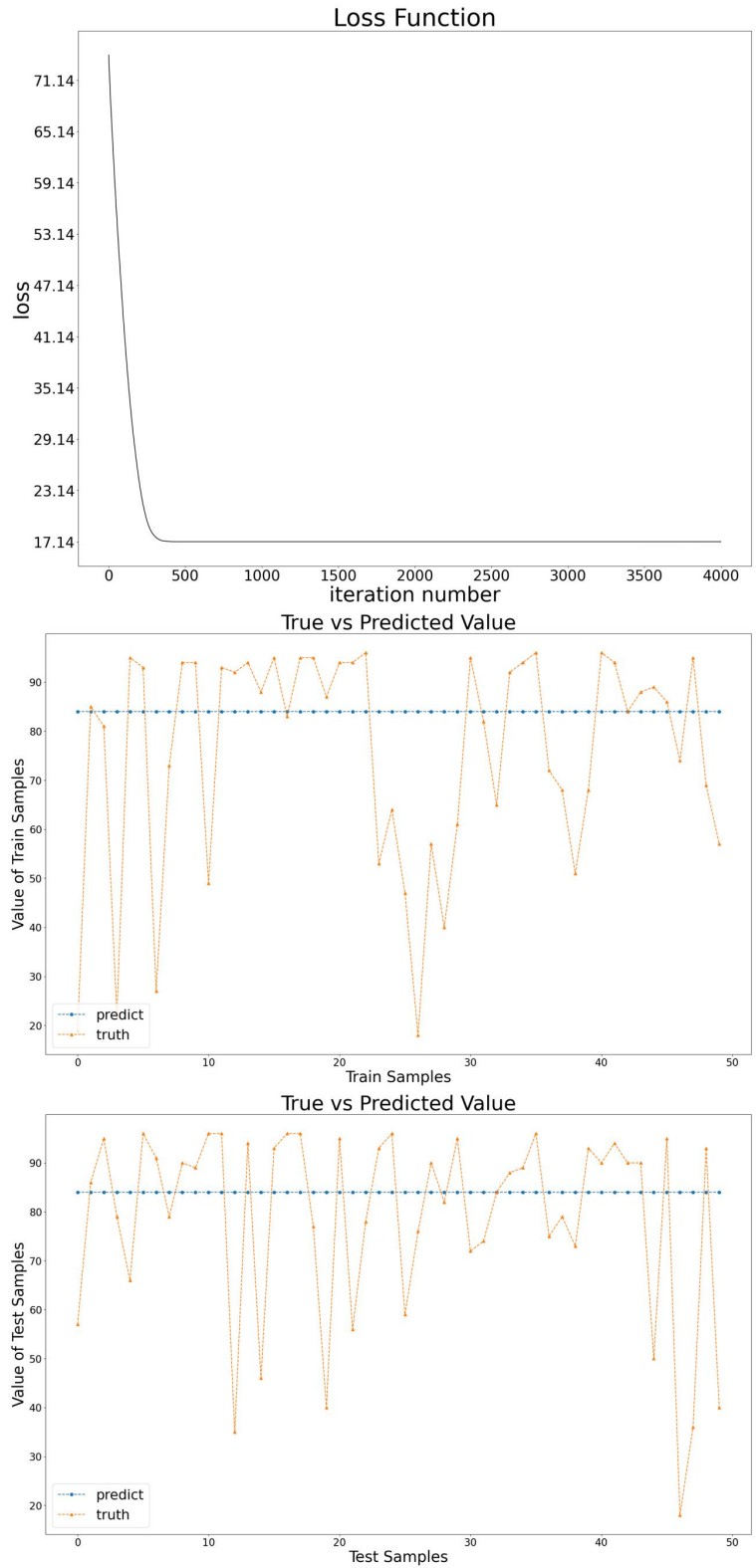

Figure 15: Learning coverage function with DSF having $\sigma(x) - 0.5$ as activation function. The used DSF has 3 layers each having 64 neurons and $\sigma(x) - 0.5$ as activation function. We can see that it still outputs constant when learning coverage function with universe size of 100 and probability of 0.5.

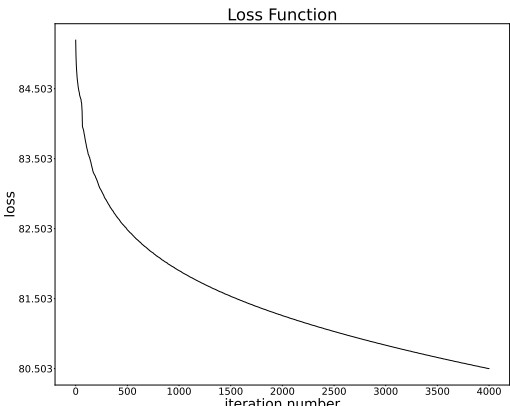

Figure 16: Loss function of EDSF with $\log(1+x)$ as activation function. The target functions is a coverage function with 100 universe size, probability of 0.2, and 31 items.

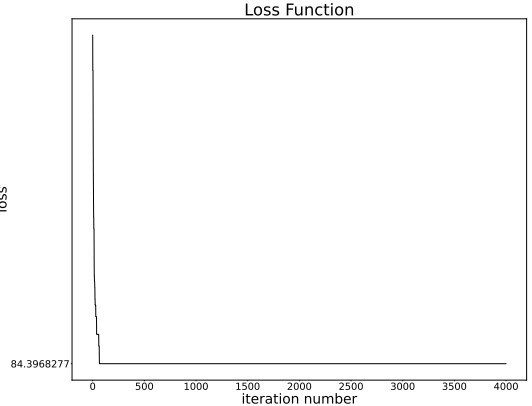

Figure 17: Loss function of EDSF with $\tanh(x)$ as activation function. The target functions is a coverage function with 100 universe size, probability of 0.2, and 31 items.

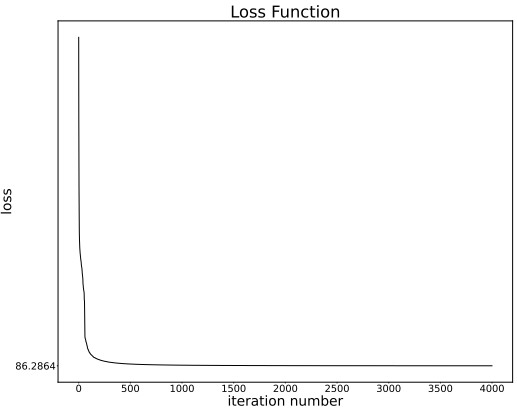

Figure 18: Loss function of EDSF with $\sigma(x) - 0.5$ as activation function. The target functions is a coverage function with 100 universe size, probability of 0.2, and 31 items.

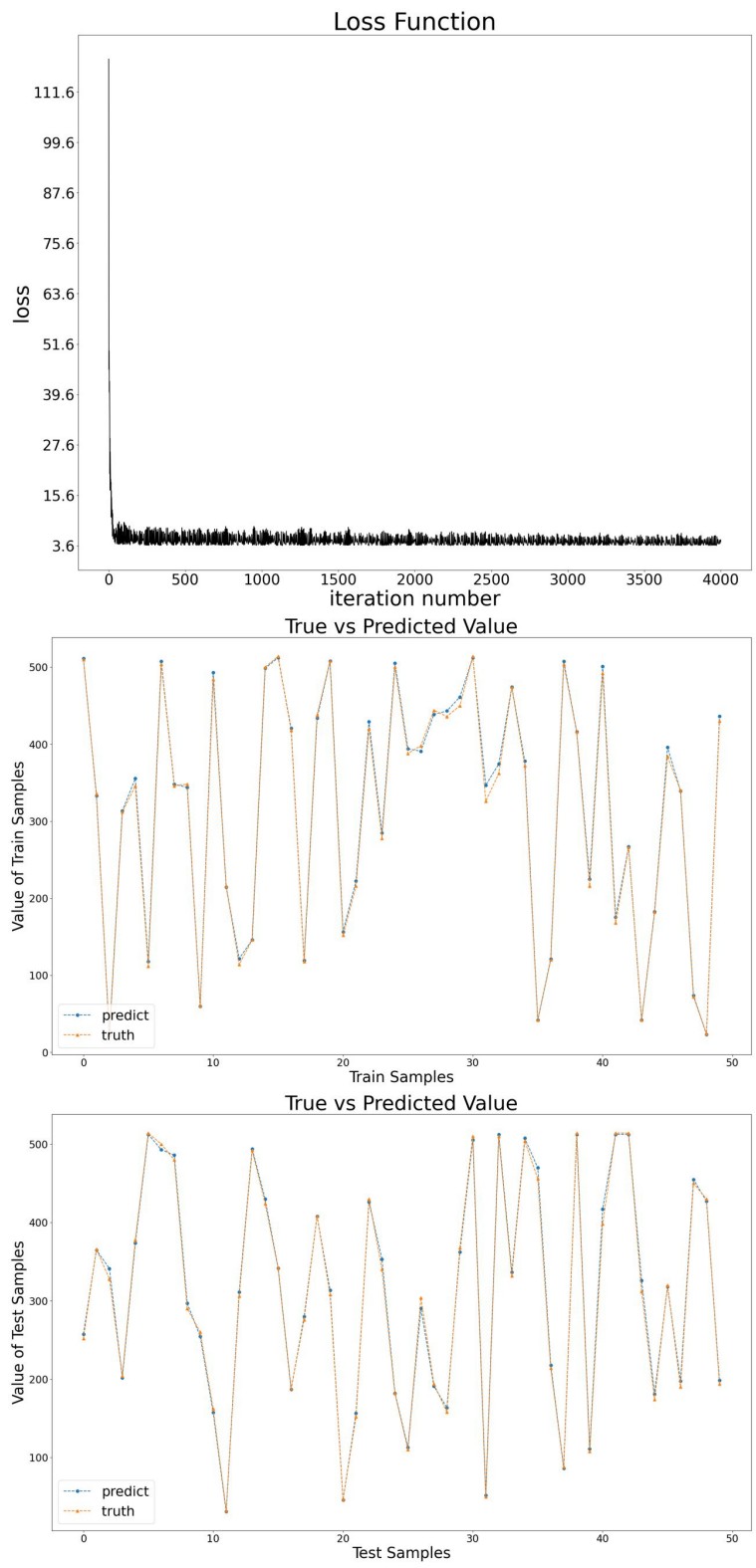

Figure 19: Learning a cut function generated by the Erdos-Renyi model with probability 0.2 and having 50 vertices, using the EDSF architecture same as Table 3, showing Training loss, Truth vs. Predicted values for train and test samples.

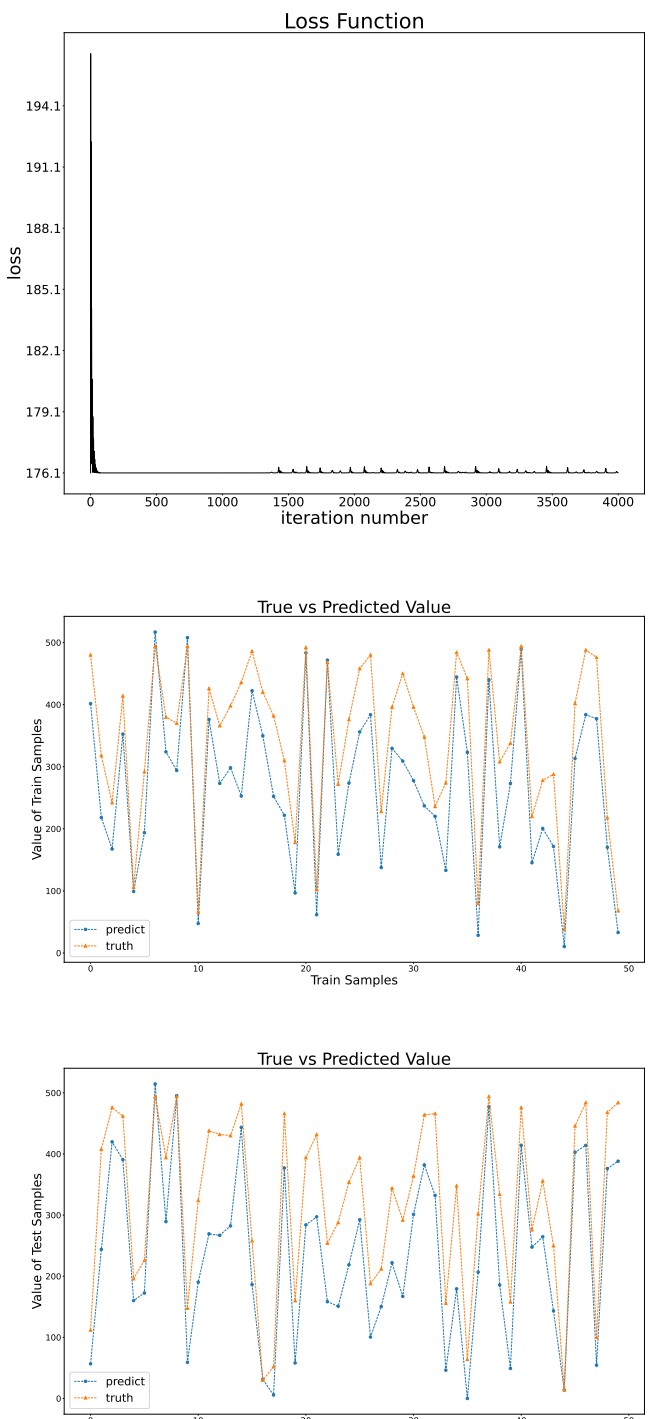

Figure 20: Learning a cut function generated by the Erdos-Renyi model with probability 0.2 and having 50 vertices, using the DSF architecture same as Table 3, showing Training loss, Truth vs. Predicted values for train and test samples.

