# OpenReview forum: "Extended Deep Submodular Functions"
_TMLR — Accepted by TMLR_

### Review · Reviewer_vKkt · 2024-10-06

**Summary Of Contributions:**

The paper presents Extended Deep Submodular Functions (ESDFs), which, as the name suggests, extend on the previous research on Deep Submodular Functions (DSFs). The paper is motivated by the fact that DSFs can not represent all possible submodular set functions and the suggested changes are heavily grounded in set theory -- EDSFs can provably represent any monotone set function. In short, an EDSF is the minimum of $r$ (a hyperparameter) DSFs.

EDSFs are evaluated on modelling coverage functions, modified graph cut functions and social welfare maximisation and the main baseline is DSFs. For all three tasks, across several settings, the proposed approach seems to outperform the baseline. Note, however, that (if I understand correctly) a separate training/optimisation is needed for each of the social welfare maximisation instances. For social welfare also a vanilla neural network is used as a baseline, but no details of what a vanilla neural network is is given.

**Audience:**

Yes

**Claims And Evidence:**

No

**Requested Changes:**

* Improve readability -- consider adding more examples/visualisations for non-straightforward mathematical concepts. (Feel free to drop figures 1-3) Note, that while this would certainly improve the paper, it is not my key issue with this work. If this part is improved, in addition to the point below, the paper may be leaning towards ["Featured certification"](https://jmlr.org/tmlr/editorial-policies.html#certifications)
* Add more technical details/discussions, as per the weaknesses outlined above. As TMLR puts an emphasis on accurate claims/rigorous evidence, this is a key aspect that has to be improved, so that I can lean for acceptance of the paper.
* Consider adding more baselines, e.g. DeepSets
* Consider open-sourcing the paper's code -- this is something I will not penalise for, if not done, but something that would aid the claims of the paper.

**Strengths And Weaknesses:**

## Strengths

* The paper is well-motivated, both in terms of what's missing in DSFs and in terms of why submodular functions are important to be modelled.
* While it can be a double-edged sword, cf. below, the paper is heavily grounded in theory, and the proposed approach is proven to be able to represent any monotone set function.
* Real-world experiments with combinatorial optimisation problems are presented.
* The paper, especially if more technical details and code are provided, would be interesting to the ML/DL community.

## Weaknesses
* The paper and its proofs are too hard to follow, at least for someone not familiar with submodular set theory. Any visualisations present are... useless. Figure 1 is more or less an MLP (something there are plenty of images on the web), and no emphasis in the figure is put on what is the difference to an MLP. Similarly, in Figure 2 -- the concepts visualised are fairly straightforward for a non-mathematical person like me... In Figure 3, I think a min-component is something obvious to people who do machine learning/deep learning, which are the readers of TMLR.

   $ $

  At the same time, definitions like "Polymatroid", "Coverage function", or Lemmas and Theorems from 3.5 onwards (3.5-3.11) are thrown in with little to no intuition/visualisation/small concrete examples. I cannot say for other readers, but improving on this would make it more readable from my POV.
* A lot of technical details are missing, limiting the reproducibility/reusability of the work. I list those that I immediately notice and I aim to publish my review as early as possible so that we can work together on detailing the paper enough so that it is reproducible
  * I could not find what the learning the learning rates are.
  * What are the dimensionalities of the hidden layers in the DSF modules and what is $r$ in Table 1?
  * For min-cut how is the neural model trained (loss functions, etc.)? Also, while the reviewer is referred to Appendix B, I couldn't immediately notice what DSFs performance is.
  * The same goes for social welfare maximisation:
     *  I understand we assume that the value functions from (27) are coverage functions, but do we use a trained EDSF model from section 4.1 to predict their values, or do we use (27) as an unsupervised loss function?
     * Table 4 states, in the caption, that the universe size is 50 and the number of items are 8, but is that 1 social welfare maximisation per experiment?
     * If the above is true, apart from it, are there any other details between experiments? (Weight initialisation seed, etc.)
     * Why, in Tables 4 and 5 (and 6 as well), do we have different optimal social welfare? Do the test instances differ? While the performance improvement will most likely remain, a scientifically rigorous experiment should test two models on the *same* data, so as to claim that one is superior to the other
     * Table 6 reports the performance of a vanilla neural network -- what is a vanilla neural network? MLP? DeepSet?
     * What is the average in Table 6 -- is it better than DSFs?
     * Weirdly, why is the optimal social welfare rarely exceeding 105, whereas in Tables 4-5 (which, according to the caption, is the same data distribution), it is always above 120. Even if datasets are different, I find this odd, given they should be the same distribution.
  * In the appendix, the loss curves, including for EDSFs are far from smooth and are formed of spikes and plateaus. Take, for example, Figure 4, top left. The loss function does not drop below 7 up until epoch 4000, then a spike occurs, and the loss drops below 7, plateauing at around 1 but never reaching 0. I cannot help but wonder why this happens -- vanishing/missing gradients due to the usage of `min`? Poor hyperparameter choice/implementation of the experiments? Or something else altogether -- I do observe that the loss of DSFs, while it plateaus at different values, suffers from the same issues. A discussion on what is going on may be useful here.
* A baseline that I believe should be present for all experiments, is the DeepSet architecture.

* No code is provided. I do not have access to a model that I could potentially reuse or code that would help me understand the experimental setting.

---

> ### Author Response · Authors · 2024-10-29
> **Response to Reviewer vKkt**
>
> We provided the code and newer version of the paper in the following links:
>
> http://195.248.243.36/paper.pdf
>
> http://195.248.243.36/comb-auction-tmlr.zip
>
> **Requested Changes**:
>
> *(Change 1)*:
> We have applied several improvements to the figures and added some examples to enhance readability in the new version of the paper. In summary, we removed Figures 1 and 3 from the old version and we have added Figures 2 and 3 (in the new version) to illustrate polymatroids and coverage functions in the paper with some examples (we have added Examples 2.8 and 2.12). Furthermore, we have added a new figure (Figure 1 in the new version) to illustrate SCMMs. Finally, we have also updated a new version of the figure that demonstrates the function $g_A$ (Figure 2 in the old version and 4 in the new version).
>
> *(Change 2)*:
>
> * *(Weakness 1)*:
> We would like to thank the reviewer for this comment. We have removed non-informative figures (as described in Change 1) and added or updated some new figures. Also, we have provided a few examples to clarify the mathematical concepts introduced in the paper.

---

> ### Author Response · Authors · 2024-10-29
> **Requested Changes 2**
>
> * *(Weakness 2)*: To address this comment, we have provided the code and the implementation details. Also, we have added the missing details of the experiments in the new version of the paper (please refer to Section 4 and Appendices A and B).
>   * In particular, the learning rate for learning coverage and cut function with both DSF and EDSF, was 0.01 and the learning rate for the gradient ascent to maximize the social welfare was 0.001.
>   * Some of the details of the structure for Table 1 were already available in Appendix A in the old version. However, in the new version, we have brought this information in the body of Section 4. Moreover, further details of the experiments have been added to Appendices A and B in the new version. For this experiment, we have employed 3 layers each having 64 neurons for both DSF and EDSF structure. As a result, r is 64 for EDSF (having 64 DSFs before the min-component).
>   * For the cut experiment, we have also employed 3 layers each having 64 neurons for both DSF and EDSF structure. As a result, r is 64 for EDSF (having 64 DSFs before the min-component). Most of the settings of the experiments are the same as the ones for the coverage function. In Table 3 (in the new version), we have also added more details of the experiment (also, refer to Section 4.2 for more details).
>   * The loss function and the learning rate for social welfare maximization are the same as training coverage functions, i.e., L1 Loss, and 0.01. We will add more details about this experiment in the newer version of the paper in Section 4.3.
>   For the social welfare maximization problem, the details of the experiment are as follows:
>     * First, we trained an EDSF separately to learn each bidder’s value function, and using these trained models we could predict the value functions for each bidder and then computed the maximizing allocation for social welfare using the gradient ascent algorithm.
>     * Yes, each row of the table is a separate experiment with different data (different coverage functions).
>     * We initialize the weights using The Gaussian distribution with low variance. We have added more details in the newer version of the paper.
>     * We first conducted a set of separate experiments to find out the overall performance of DSFs and EDSFs in the estimation of maximizing social welfare (Tables 4 and 5 in the new version). Next, we conducted another set of experiments on the same data to compare them fairly in practice which is available in Tables 7 and 8 (in the new version) with varied settings. In these tables, in each experiment, the two DSF and EDSF architectures are compared on the same problem instances.
>     * The vanilla neural network has two differences from our proposed models. First, the weights could be negative and the activation functions could be non-concave. We have used the ReLU activation function as a baseline. We have added more details about the experiment in the newer version of the paper (see Table 6).
>     * We have added the missing average values to the table. Also, as observed by the reviewer, the overall performance of the vanilla neural networks in our experiments is better than the DSF’s performance.
>     * Thanks for mentioning this point, there is a typo in the caption about the universe size of the coverage function in Tables 4 and 5 (The universe size is 60 instead of 50). We have corrected it and added additional details in the newer version of the paper.
>   * We have added a discussion about this behavior of the loss function to the discussion section in the newer version of the paper (see Remark 5.4). However, in our experiments, we found out that some factors have effects on the frequency and magnitude of the fluctuations. The most important one is the activation function MiLU, having sharp edges and a tendency to become saturated. This behavior of MiLU activation makes the learning process much harder. To test our conjecture, we ran multiple new experiments with EDSFs having smoother activation functions (like log(1+x), refer to Appendix A). It can be seen in the new plots (Figures 16-18), that the loss functions are much smoother and have no visible fluctuations. It's worth noting that EDSFs with these activation functions are not as well as the MiLU activation function when learning the coverage function.
>
> * *(Weakness 3)*: To address this comment, in the new version of the manuscript, we have added two new baselines, namely, Deep Set and Set Transformers, and compare the proposed method against them (refer to Tables 1 and 3 in the new version).

---

### Review · Reviewer_2jd5 · 2024-10-11

**Summary Of Contributions:**

The paper introduces a class of set functions representable as neural networks: Extended Deep Submodular functions (EDSFs). This class is an extension of the Deep Submodular Functions (DSFs), which is a class of submodular functions representable as neural networks that were introduced by (Bilmes & Bai, 2017). An EDSF is defined as the minimum of r DSFs. The authors show that the class of EDSFs is equal to the class of monotone set functions. In particular,
they show that any monotone set function can be represented by an EDSF with 3 layers and exponentially many nodes in the first hidden layer. They provide some very small scale empirical results on learning coverage functions on synthetic datasets showing better performance when using EDSFs compared to DSFs.

**Audience:**

Yes

**Broader Impact Concerns:**

None.

**Claims And Evidence:**

No

**Requested Changes:**

Questions:
- What does it mean to vary r (Table 2) in terms of the architecture used? In appendix A, you state that you use three fully-connected layers with 64 neurons. I am guessing that means $r = 64$ in this case. When you vary r, do you only vary the number of neurons in the last hidden layer, or in all layers?
- What is the coverage probability used in Table 2? Shouldn't the test loss for $r = 64$ match one of the test losses in Table 1?
- Did you test using non-uniform $\alpha$ values? I would expect this to perform better.

Critical requested changes:
- Add missing related work
- To support the claim that EDSFs outperform DSFs, other architecture choices for DSFs should be explored, for example different choices of concave functions.
- Either adjust/remove claims like "EDSFs present a promising advancement in the representation and learning of set functions with improved generalization capabilities", or add large scale non-synthetic experiments and compare with other related work to properly support this claim (see non-critical requested changes below).
- Since the result that any monotone function can be represented as a EDSF (Theorem 3.11) implies the result for monotone submodular function (Theorem 3.9). Either remove that result, or explain why it is interesting.
- Theorem 3.9 can be proved in the following much simpler way, which does not require going through polymatroids:
The submodular function $g_A(B)$ corresponds to $F(A) + \sum_\{j \in B \setminus A\} F(j)$ if $A$ intersects $B$, and $\sum_\{j \in B\} F(j)$ otherwise. Hence, any monotone submodular function $F$ satisfies $F(B) \leq \min_A g_A(B)$ (see NemhauserAnalysisApproxSubMax-II Prop 1.1-(iii)). It directly follows then that $F(B) = \min_A g_A(B)$, since $g_B(B) = F(B)$.
If you decide to keep this theorem, replace its overly lengthy proof by this simpler proof, and remove claims that this result requires "an analysis of polymatroid properties".
- In Theorem 3.2, remove "(submodular)" from the statement, it is confusing.
- In the definition of SCMM (Definition 2.5), the arbitrary modular function m± should take any real value, not just non-negative. If you want to consider only monotone SCMM, this should be explicitly stated.
- In the definition of DSF (Definition 2.6), there should be an arbitrary modular m± added at the end. DSFs are not necessarily monotone! If you want to consider only monotone DSFs, this should be explicitly stated.
- Include experimental setup details such as: optimization method and parameters used for training, how the number of neurons and $\alpha$ parameters were chosen, how the coverage functions for valuation functions are generated in section 4.3., what values of $n$ and $\eta$ you use?
- Include variance values for all results (these are missing in Table 1 and 2)
- In tables 4-8, include both the optimal solutions for both the learned functions and the true underlying ones.
- Modify the claim that you "introduce a novel category of set functions" to say "novel representation of monotone set functions", since EDSFs corresponds to monotone set functions.
- In the discussion section, the complexity of computing the gradient in GA is incorrect; it's not a constant! Note also that faster variants of continuous greedy are given in [Mokhtari et al, 2020] and [Badanidiyuru et al, 2014].

Mokhtari, A.; Hassani, H. & Karbasi, A.
Stochastic conditional gradient methods: From convex minimization to submodular maximization
The Journal of Machine Learning Research, JMLRORG, 2020

Badanidiyuru, A.; Mirzasoleiman, B.; Karbasi, A. & Krause, A.
Streaming submodular maximization: Massive data summarization on the fly
Proceedings of the 20th ACM SIGKDD international conference on Knowledge discovery and data mining, 2014

Non-critical requested changes:
- Adding more large scale non-synthetic experiments and comparing with other related work is needed to really demonstrate the feasibility and advantage of EDSFs. In particular, I suggest to compare with the approach of (De & Chakrabarti, 2022), Deep sets (Zaheer et al, 2017), and Set Transformer (Lee et al, 2019).
- the name "extended deep submodular functions" is confusing since the functions in this class are *not* necessarily submodular!
- release the code for reproducibility
Typos:
- In the proof of Lemma 3.5, the first equality in the 2nd case should be an inequality.

**Strengths And Weaknesses:**

Strengths:
- Paper introduces a novel representation of monotone set functions as neural networks.
- The provided limited empirical results show that learning coverage functions with EDSFs seem to yield significantly better performance compared to DSFs.
- The paper is written clearly.

Weaknesses:
- The proposed neural network representation has exponential width. So it is not possible to provably learn it in polynomial time.
- The theoretical results provided are straightforward observations (see requested changes for more details).
- Experimental results are very limited:
	- the datasets used are very small (largest ground set size considered is 50 for the learning experiments, and 8 for the submodular welfare maximization) and synthetic.
	- only coverage and cut functions are considered, which are both monotone submodular. Since the introduced class can represent any monotone set function, it would be good to test performance on a non-submodular function.
	- they only compare their approach with the DSF approach from [Bilmes & Bai, 2017]. And they only consider one type of activation function (minimum linear unit), which might be more suitable for EDSF.

- A lot of details related to their experimental setup are missing (see requested changes).

- Several related work are not discussed, such as:
	- Theoretical work on learning general submodular functions:

	Vitaly Feldman and Jan Vondrak. Optimal bounds on approximation of submodular and xos functions by juntas. SIAM Journal on Computing, 45(3):1129–1170, 2016.

	- learning mixture of fixed submodular functions:

	H. Lin and J. Bilmes. Learning mixtures of submodular shells with application to document summarization. In Uncertainty in Artificial Intelligence (UAI), Catalina Island, USA, July 2012. AUAI.

	R. Sipos, P. Shivaswamy, and T. Joachims. Large-margin learning of submodular summarization models. In Proceedings of the 13th Conference of the European Chapter of the Association for Computational Linguistics, pages 224–233. Association for Computational Linguistics, 2012.

	- deep set functions:

        Juho Lee, Yoonho Lee, Jungtaek Kim, Adam Kosiorek, Seungjin Choi, and Yee Whye Teh. Set transformer: A framework for attention-based permutation-invariant neural networks. In ICML, 2019.

	Edward Wagstaff, Fabian Fuchs, Martin Engelcke, Ingmar Posner, and Michael A. Osborne. On the limitations of representing functions on sets. In Proceedings of the 36th International Conference on Machine Learning, 2019.

	- decision-focused learning:

	Sakaue, Shinsaku. "Differentiable greedy algorithm for monotone submodular maximization: Guarantees, gradient estimators, and applications." International Conference on Artificial Intelligence and Statistics. PMLR, 2021.

	B. Wilder, B. Dilkina, and M. Tambe. Melding the data-decisions pipeline: Decision-focused learning for combinatorial optimization. In Proceedings of the 33rd AAAI Conference on Artificial Intelligence, pages 1658–1665. AAAI Press, 2019.

---

> ### Author Response · Authors · 2024-10-29
> **Reponse to Reviewer 2jd5**
>
> We have provided the newer version of the paper and the code in these links:
>
> http://195.248.243.36/paper.pdf
>
> http://195.248.243.36/comb-auction-tmlr.zip
>
> **Requested Changes**:
>
> **Questions**:
>
> *(Question 1)*
> $r$ means the number of DSFs that are used before the min-component in the EDSF structure. So when we vary $r$ it means that the number of neurons in the last layer is varied and the number of layers for the previous layers was not changed (remained 64 throughout the experiments).
> In the setting we used for our experimentation, all layers have the same number of neurons for simplicity. However, the number of neurons for the last layer is theoretically more important and shows the number of DSFs used in the EDSF structure. So we only needed 64 DSFs in practice to estimate the coverage function properly.
>
> *(Question 2)*
> The probability of the coverage function in Table 2 (in the current version) is 0.2.
> The experiments’ setup of Tables 1 and 2 (in the previous version) was different. So to have concise results, we have rerun the experiments for a new setup, as can be seen in Tables 1 and 2 (in the new version of the paper).
>
> *(Question 3)*
> We have tested different values for alphas including non-uniform ones for different layers of DSFs. However, none of them did work and had the same problem of outputting constant value. It is worth mentioning that finding the best alpha in the DSF setup (and generally setting the hyperparameters, including the structure of the DSF and alphas) is a challenging task. In contrast, different structures and alphas for EDSF did work properly in practice which made finding the hyperparameters a much easier task.

---

> > ### Author Response · Authors · 2024-10-29
> > **Critical Changes 1**
> >
> > **Critical Requested Changes**:
> >
> > *(Change 1)*:
> > In the current version, we have added the related works mentioned by the reviewer, which can be found in Section 1.1 in blue color.
> >
> > *(Change 2)*:
> > We did try different structures (number of layers and neurons) and activation functions but none of them worked properly and outputted near-constant results. Refer to Figures 11-15 in the new version of the paper.
> > The result for $\log(1+x)$, $\tanh(x)$, and $\sigma(x) - 0.5$ is added in the new version (Figures 13-15).
> > In addition to investigating various activation functions, and in response to the reviewer Ythe, we have also performed experiments with the following setups (Refer to Figures 11, and 12 for more details):
> > * 4 layers with 2048 neurons each with MiLU activation function,
> > * 6 layers with 2048 neurons.
> >
> > It’s also worth noting that there is a theorem that states that each concave activation function can be computed using a (probably infinite) number of MiLU activation functions [Bilmes 2017, Theorem 5.9]. So it seems that the MiLU activation function can cover all the concave activation functions and is rich enough.
> >
> > *(Change 3)*:
> > We have rewritten the abstract to make the claim more precise and removed the mentioned sentence by the reviewer. In addition, following the reviewer’s suggestion, we have performed several new experiments to compare EDSF’s performance with Deep Sets and Set Transformers, which have been added in the revised manuscript (Sections 4.1 and 4.2). Moreover, we conducted extensive searches to find a large-scale, non-synthetic dataset to test the performance of the EDSF in learning. However, as far as we could determine, there was only one dataset available (the Amazon Baby Registry), and we did not find any well-established documentation for it.
> >
> > *(Change 4)*:
> > We thank the reviewer for the careful reading of our paper and the technical materials. We would like to note that the architecture of EDSFs was achieved through an analysis of polymatroids, which also formed the basis for proving Theorem 3.9. We believe that including Theorem 3.9, along with its proof based on polymatroid theory, is essential for providing a better understanding of EDSFs. To emphasize the importance of Theorem 3.9, we have added a remark in Section 5 (Discussion and Remarks).
> >
> > *(Change 5)*:
> > As we mentioned in the previous requested change, we believe that the proof of Theorem 3.9 through the analysis of polymatroid provides a better intuition about the construction of EDSFs.
> > However, for the sake of completeness, we have presented the simpler proof of Theorem 3.9, kindly provided by the reviewer, in Appendix C (in the new version).
> >
> > *(Change 6)*:
> > The requested change is applied.
> >
> > *(Change 7)*:
> > Yes, we wanted to consider only monotone set functions. The change is done in the new version of the paper.
> >
> > *(Change 8)*:
> > Yes, same as the previous comment, we want to only consider monotone set functions. To emphasize this fact, we have added Footnote 1 in the new version.
> >
> > *(Change 9)*:
> > The optimization method used to train the neural networks was Adam and the learning rate was 0.01 for learning coverage and cut function. The loss function was L1-loss. The number of iterations (also epochs) varied between 4000 to 10000.
> > For setting hyperparameters like the number of layers, neurons, and the value for alpha, we did an exhaustive search and tried different nominees for the number of neurons, number of layers, and the value for alpha. Although for EDSF many different hyperparameters worked well, for DSFs none of our nominees had a promising output or loss.
> > In section 4.3, n is 3 and eta is 0.001.
> > More details about the hyper-parameters are included in the new version of the paper in Section 4 and appendices A and B.
> >
> > *(Change 10)*:
> > For all the results presented in Tables 1-3 in the new version, we have added the standard deviation for each experiment.

---

> > > ### Author Response · Authors · 2024-10-29
> > > **Critical Changes 2 & Non-critical Changes**
> > >
> > > *(Change 11)*:
> > > To address this comment, we have added a new table to the manuscript (Table 9), which contains two different optimal values (the optimal solutions for both the learned functions and the true underlying ones) for DSFs and EDSFs architectures.
> > >
> > > *(Change 12)*:
> > > The requested change has been made.
> > >
> > > *(Change 13)*:
> > > * Regarding the complexity of GA, by O(1) we meant that without considering the input layer, we would have a neural net with some weights having no dependency on n and m, resulting in a constant order of each iteration of backpropagation.
> > > * However, if we consider our proposed architecture, the size of the network is exponential and it is not informative to compare it with the other related methods.
> > > * The main point we would like to mention in the paper is to compare the performance of the gradient ascent (GA) algorithm with the related algorithms, like Randomized Greedy (RG), in finding the maximum of the social welfare “in practice”. However, to address the reviewer’s comment, we have modified this remark in the new version of the paper (Remark 5.3) to be more clear. In the current version, we have practically compared the performance of the proposed method against the RG algorithm, and we have reported the runtime of both methods. As can be observed, our proposed method is about 5-6 times faster than RG.
> > > * In comparison with other mentioned stochastic continuous greedy algorithms, we would like to mention that there is work done in [Bilmes, Bai, 2018, Neurips], which has a better approximation factor and iteration number when the number of layers in DSFs is sufficiently large. Going through their proof, we believe that a similar result can also be extended to EDSFs. If this is true, it shows that the proposed method has a better approximation than RG, and the two other works mentioned by the reviewer, if the number of layers is large enough.
> > >
> > > **Non Critical Requested Changes**:
> > >
> > > *(Change 1)*:
> > > To address this comment, we have added the results of Deep Set and Set Transformer as two baselines to compare with the proposed method (also please refer to the response to Change 3). Moreover, as far as we searched, we could not find any well-documented non-synthetic dataset.
> > >
> > > *(Change 2)*:
> > > Thank you for mentioning this comment. Since the proposed method is rooted in DSF and is an extension of it, we have used this name for the proposed architecture. We will consider this issue for the further improvement of the paper if it is allowed by the journal policy.
> > >
> > > *(Change 3)*:
> > > The code is released:
> > >
> > > *(Change 4)*:
> > > Thanks for the comment. We have fixed it.

---

> > > > ### Comment · Reviewer_2jd5 · 2024-11-08
> > > >
> > > > Thank you for making most of my requested changes.
> > > >
> > > > I have a few follow-up comments regarding (change 13):
> > > > - The algorithm from (Vondrak 2008) is called "Continuous Greedy" or "Randomized Continuous Greedy". Avoid referring to it as "Randomized Greedy" as this is confusing. There exists a different algorithm with that name in the literature. The algorithm from Vondrak is usually referred to as "Continuous Greedy".
> > > > - Note that it is possible to obtain better approximation guarantee for general DSFs as done in [Bilmes, Bai, 2018, Neurips] because this is a subclass of submodular functions. However for general submodular functions (and thus also submodular EDSFs) this is not possible, as (1-1/e) approximation is provably optimal in that case. It might be possible to extend the result of [Bilmes, Bai, 2018, Neurips] to a subclass of EDSFs (probably if you restrict the number r of DSFs therein to be polynomial).
> > > > - It is important to at least acknowledge in the paper that faster variants of continuous greedy are given in [Mokhtari et al, 2020] and [Badanidiyuru et al, 2014], when discussing the time complexity of continuous greedy. Also the 2nd limitation stated for CG is somewhat misleading: CG like GA can be applied to non-submodular functions. If by applicable you meant having provable approximation guarantee. Then GA is also only guaranteed to optimally solve the relaxed problem for non-submodular functions that can be written as EDSF with a polynomial number r of DSFs.
> > > >
> > > > I also have the following suggestions for improving the clarity of Theorem 3.12:
> > > > - Specify where in [Bilmes & Bai, 2017] is the result cited on DSFs being concave over non-negative inputs.
> > > > - It is worth explicitly stating that when optimizing set functions, the input to the EDSFs is a binary vector, so the function is concave in that case.

---

> ### Author Response · Authors · 2024-11-16
> **Reply to Reviewer 2jd5**
>
> Thanks for your comment.
>
> Updated version of the paper: http://195.248.243.36/TMLR_Journal_Submissions-v2.pdf
>
> * This change has been made in the final version of the paper.
> * We thank you for this clarifying comment. We still believe that the proofs from [Bilmes, Bai, 2018, Neurips] can be directly applied to the EDSF setting, as for each input $[0,1]^n$, only one of the DSFs is active (the DSF with the minimum output value). However, this does not contradict with the stated fact, since the proposed architecture will be exponential in input size. We would like to note that we had not put this claim in the paper.
> * This change has been made in the final version of the paper and we have acknowledged this fact in Remark 5.3. We have also removed the second limitation of the CG algorithm stated in Remark 5.3.
>
> * It is cited in [Bilmes, Bai, 2018, Neurips][Corollary 1]. We have fixed it in the final version.
> * Thank you for your comment, we have added the explanation in a footnote in the p11 in the final version.

---

> > ### Comment · Reviewer_2jd5 · 2024-11-16
> >
> > Thanks for your response and implementing these changes. Please add "Corollary 1" when citing [Bilmes, Bai, 2018]. It is helpful to point the reader to where the result can be found in the reference.

---

### Review · Reviewer_Ythe · 2024-10-15

**Summary Of Contributions:**

The paper introduces a novel class of set functions called Extended Deep Submodular Functions (EDSFs). The idea is super simple yet nice, the min function over a set of DSFs, which significantly extend Deep Submodular Functions (DSFs) by being capable of representing all monotone submodular functions. The authors demonstrate through theoretical analysis and experiments that EDSFs generalize much better than DSFs in learning complex submodular functions.

The paper is well written and inspiring. Overall, I enjoy reading the submitted manuscript.

**Audience:**

Yes

**Claims And Evidence:**

Yes

**Requested Changes:**

Here I have some questions regarding the empirical experiments:

1. Learning coverage functions: It seems that DSFs are always outputting constants. I’m interested in the expressive power of DSFs when representing the coverage functions. It would be great if two following experiments are conducted:

1.1 For fairness use the same number of neurons as EDSF (e.g. like 64*3*3).

1.2 Try the boundary of DSF (e.g. a super larger DSF), see its capability in representing coverage functions.

2. Learning cut functions: I went throught appendix. B, but found no results of DSFs in learning the cut functions. Even though the DSF performs reasonably good, the authors should give details of such result and explain the potential reason.

3. Learning social welfare maximization:

3.1 It’s shown that DSFs could not probably learn coverage functions well, yet here the authors presume the true submodular functions are coverage functions. I suggest that the authors may change another presumption for fairness.

3.2 Does Vanilla NN have same size (number of neurons) as EDSF?


[minor] it would be rigorous to add std in Table. 1.

**Strengths And Weaknesses:**

**strength:**

1. EDSFs can represent all monotone submodular and set functions. This addresses a major limitation of DSFs, enhancing the modeling of submodular functions in ML.

2. Experiments provide clear evidence that EDSFs outperform DSFs, (empirical generalization error).

**weakness:**

One significant drawback is the potential exponential growth in the network size needed for the expressiveness. Although fewer DSFs can work in practice, this complexity might limit scalability and hinder practical implementation for larger datasets (or say, real-world problems). But the paper also noted this in the conclusion.

I have some questions regarding the empirical results. Please refer to the requested changes.

---

> ### Author Response · Authors · 2024-10-29
> **Responses to Reviewer Ythe**
>
> We have added the newer version of the paper and the code in these links:
>
> http://195.248.243.36/paper.pdf
>
> http://195.248.243.36/comb-auction-tmlr.zip
>
> 1.1. We would like to note that the same number of neurons is already used in the paper for both DSF and EDSF architectures when we compare them. Most of the results shown in the paper are done with the same structure for both DSFs and EDSFs (they only differ in the last neuron that in DSF architecture sums its input but in the EDSF architecture finds the min value of its inputs).
>
> 1.2. We tried many different sizes and architectures of DSFs including the very small ones and larger ones. All large DSFs have the problem of outputting constant value and the smaller ones cannot estimate the target function properly with low error (refer to Figures 10-16 in the new version). In contrast, setting the hyperparameters and architecture of EDSFs is much easier and actually as supported by the theory behind EDSFs, adding more neurons to the EDSFs architecture and making it wider adds to the expressibility of the EDSFs.
>
> 2. We have added the results of DSF in the new version of the paper (Section 4.2 and Appendix B). We can observe that DSF can identify and follow the pattern of target cut function, but not as well as the EDSF with larger test and train loss. It is worth mentioning that to find the best hyperparameters for DSFs we tried many different numbers of layers, neurons, and values for alpha. Many of them had the problem of outputting constant values. Still, a few of them (including the one mentioned in section 4.2 in the newer version of the paper) did not have this problem and performed reasonably better. The reason behind the better performance for DSFs in learning the cut function in comparison with the coverage function could be that the cut function is simpler than the coverage function to learn. However, as we mentioned before, few sets of hyper-parameters for DSFs did work but for EDSFs, many sets of hyper-parameters worked well.
>
> 3.
>
>   3.1.
> We wanted to show that social welfare maximization does not work well because of poor learning of the true value functions. If DSF could learn the value function properly (with low error, like another function than the coverage function) the gradient ascent algorithm can also work with DSFs because they have the concavity property like EDSFs. The results were shown as a baseline for comparison with EDSF results.
>
> We have performed a new experiment about social welfare maximization when the ground truth value function is DSF (large DSFs with four layers of size 128 neurons each). As mentioned above, when the DSFs are good at learning, the social welfare is near the optimal since the DSF functions are concave, and the gradient ascent works well to estimate the optimal value:
>
> | EDSF Social Welfare | DSF Social Welfare | Optimal Social Welfare|
> -------------------------|------------------------|---------------------------|
> |7090 | 7462 | 7462 |
>
> The reason behind the exactness of estimated social welfare and the optimal social welfare is the quality of generalization of the learning model.
>
>   3.2.
> Vanilla NN has 3 layers each having 64 neurons with ReLU activation function. So it has the quite same structure as EDSF.
> It is also good to mention that not only does vanilla NN not learn the coverage function properly, but the gradient ascent algorithm of social welfare maximization can’t work well either (because of not having the concavity property of vanilla NN).
>
>
> [Minor]:
> We have added a complete set of experiments for learning coverage and cut functions with a mean and standard deviation of the outputs of the experiment set. See Tables 1 and 3 in the newer version of the paper.

---

### Public Comment · ~Gantavya_Bhatt1 · 2024-12-18
**Related work discussion with Deep Submodular Peripteral Networks**

Hi!

Thanks for the interesting work. I'd like to point out to our paper Deep Submodular Peripteral Networks (DSPN) (https://arxiv.org/abs/2403.08199) which appeared at NeurIPS earlier this year, since our paper also extends beyond existing DSF class, including the cycle matroid rank function. I would appreciate if the authors can mention them in the related works.

Thanks!

---

### Decision · Action_Editor_GZFB · 2024-11-19

**Recommendation:** Accept with minor revision

**Comment:**

Reviewer 2jd5 has pointed out that the discussion on continuous greedy still needs revising, as pointed out in their follow-up response.  Please do so, and enter (on OpenReview) a summary of the edits.

Reviewers 2jd5 and Ythe both pointed out the strong limitation of exponential width.  I see that this is discussed in Section 5, but given how significant of a limitation it is, it should also be mentioned earlier -- probably Section 3, and maybe even in the first paragraph of Page 2.

**Audience:**

This paper is a strong match to TMLR, with both deep neural networks and submodularity being widely-considered topics in the machine learning literature.

**Claims And Evidence:**

This paper introduces Extended Deep Submodular functions, which improve over the non-extended version by being able to represent all monotone submodular functions, while maintaining desirable concavity properties.  These results are supported via mathematical theorems and proofs.  Empirical studies are also given for coverage functions, cut functions, and social welfare problems, and improved generalization properties are observed over the existing baselines.

---

> ### Author Response · Authors · 2024-12-14
> **Response to the Action Editor**
>
> The two comments raised by reviewer 2jd5 has been addressed.
>
> For the first comment please refer to p. 11, in the proof of Theorem 3.12.
>
> For the second comment please refer to p. 2, 3rd paragraph and p. 11, before Section 3.2.
>
> We have also published the code on github and refer to it in a footnote on page 2.

---

> > ### Comment · Action_Editor_GZFB · 2024-12-16
> > **Follow-up**
> >
> > Dear authors,
> >
> > I'm a bit confused about addressing the first comment, which was about Continuous Greedy -- Theorem 3.12's proof doesn't seem to be related to that?
> >
> > I spotted a typo "Corrolary" and suggest you do a spell-check on the paper.  Also I would change "using much less number of DSFs" to "using much fewer DSFs" (or alternatively "using a much smaller number of DSFs").
> >
> > Action Editor

---

> ### Author Response · Authors · 2024-12-18
> **Response to the Action Editor**
>
> We apologize for the oversight.
>
> We have now included a new discussion in the first paragraph of page 19 to compare the approximation factors of Continuous Greedy and EDSFs. We believe that this comment was the only one not addressed regarding the discussion on Continuous Greedy by Reviewer 2jd5. However, if there are any other requested changes, please let us know.
>
> Additionally, we have corrected the typos and grammar errors.